# On the similarities and differences between the products of oxidation of hydrocarbons under simulated atmospheric conditions and cool-flames.

Roland Benoit [1] Nesrine Belhadj[1,2], Maxence Lailliau[1,2], and Philippe Dagaut[1]

[1]CNRS-INSIS, ICARE, Orléans, France, roland.benoit@cnrs-orleans.fr, nesrine.belhadj@cnrs-orleans.fr, maxence.lailliau@cnrs-orleans.fr, dagaut@cnrs-orleans.fr
[2]Université d'Orléans, Orléans, France

**Correspondence**: Benoit Roland (roland.benoit@cnrs-orleans.fr)

**Abstract.**

Atmospheric oxidation chemistry, and more specifically, photooxidation, show that long-term oxidation of Organic Aerosol (OA) progressively erases the initial signature of the chemical compounds and can lead to a relatively uniform character of Oxygenated Organic Aerosol (OOA). This uniformity character observed after long reaction time seems to contrast with the great diversity of reaction mechanisms observed in the early stages of oxidation. The numerous studies carried out on the oxidation of terpenes, and more particularly on limonene for its diversity of reaction sites (endo and oxo cyclic), allow to study this evolution. We have selected for their diversity of experimental conditions, nine studies of limonene oxidation at room temperature over long reaction times to be compared to the present data set obtained at elevated temperature and short reaction time in order to investigate the similarities in terms of reaction mechanisms and chemical species formed. Here, the oxidation of limonene-oxygen-nitrogen mixtures was studied using a jet-stirred reactor at elevated temperature and atmospheric pressure. Samples of the reacting mixtures were collected and analyzed by high resolution mass spectrometry (Orbitrap) after direct injection or after separation by reverse-phase ultra-high-pressure liquid chromatography and soft ionization, i.e., (+/-) HESI and (+/-) APCI. Unexpectedly, because of diversity of experimental conditions in terms of continuous-flow tank reactor, concentration of reactants, temperature, reaction time, mass spectrometry techniques and analyses conditions, the results indicate that among the 1138 presently detected molecular formulae, many oxygenates found in earlier studies of limonene oxidation by OH and/or ozone are also produced under the present conditions. Among these molecular formulae, highly oxygenated molecules and oligomers were detected in the present work. The results are discussed in terms of reaction pathways involving the initial formation of peroxy radicals ($RO_2$), isomerization reactions yielding keto-hydroperoxides and other oxygenated intermediates and products up to $C_{25}H_{32}O_{17}$. Products which could derive from $RO_2$ autoxidation via sequential H-shift and $O_2$ addition ($C_{10}H_{14}O_{3,5,7,9,11}$) and products deriving from the oxidation of alkoxy radicals (produced by $RO_2$ self reaction or reaction with $HO_2$) through multiple H-shifts and $O_2$ additions ($C_{10}H_{14}O_{2,4,6,8,10}$). The oxidation of $RO_2$, with

possible occurrence of the Waddington mechanism and of the Korcek mechanism, involving H-shifts are also discussed. The present work demonstrates similitude between the oxidation products and oxidation pathways of limonene under simulated atmospheric conditions and in those encountered during the self-ignition of hydrocarbons at elevated temperatures. These

results complement those recently reported by Vereecken and Nozière and confirms for limonene the existence of an oxidative chemistry of the alkylperoxy radical beyond 450 K based on the H-shift (Vereecken et Nozière, 2019, 2020).

## 1. Introduction

Air particulates are responsible for increasing death rates and deseases worldwide (Lim et al., 2012). Furthermore they have a negative impact on climate (Myhre et al., 2013). With increasing temperatures, particularly in summer, biogenic emissions of

volatile organic compounds (VOCs) are more important than anthropogenic. Among them terpenes emitted by vegetation represent a large fraction of the volatile organic compounds present in the troposphere (Seinfeld and Pandis, 2006;Llusia` and Penuelas, 2000). In addition, one should note that these cyclic hydrocarbons are also considered as potential high-density biojet fuels (Pourbafrani et al., 2010;Meylemans et al., 2012;Harvey et al., 2010;Harvey et al., 2015). Their use as drop-in ground transportation fuel could also be of interest, considering their cetane number around 20 (Yanowitz et al., 2017). Their use as

fuel would likely increase their emission into the troposphere. The atmospheric oxidation kinetics of terpenes has been extensively studied, although we are far from a detailed understanding of the many processes involved (Berndt et al., 2015). Monoterpenes such as α-pinene, β-pinene, and limonene are among the most abundant terpenes in the troposphere (Witkowski and Gierczak, 2017;Zhang et al., 2018). Their oxidation can yield a large variety of oxygenated organic compounds such as highly oxygenated molecules (HOMs) which are considered to play an important role in secondary organic aerosols (SOA)

formation (Bianchi et al., 2019).

Recently, Kourtchev et al. have shown that the concentration of SOA (directly related to that of VOC) mainly influences the apparition of oligomers whereas environmental or experimental conditions (RH, ozonolysis vs. OH-oxidation/photolysis, long-term atmospheric aging) preferentially influences the evolution of these oligomers (Kourtchev et al., 2016). This initial concentration is presented as one of the determining factors of the chemical nature of SOAs and their evolution into oligomers.

It was also shown that, under different conditions of oxidative aging, most of the chemical evolution of SOA was due to the reaction of the OH radical (Kourtchev et al., 2015).

Evolution of these primary compounds during atmospheric oxidation have been widely studied in atmospheric chambers, in Potential Aerosol Mass (PAM), but also in field measurements in order to understand the oxidation processes. In the case of limonene, although the fundamental aspects of the oxidation processes are mostly known, there is a real difficulty to observe

and characterize these chemical species in the initial phase of oxidation under the above conditions. This difficulty is notably reinforced, in the first steps, by the isomerization phenomena present during the oxidation of alkylperoxy radicals, $RO_2$ and the instability of these chemical species. These reaction mechanisms are mainly related to the H-shift whose rate constant

varies with temperature, but may also be dependent on the presence of double bonds (Nozière et Vereecken, 2019). In atmospheric chemistry, hydrogen migration on carbon chains is a critical step in the formation of highly oxygenated molecules.

It has been recently shown that this mechanism, which is also responsible for the formation of OH radicals, increases with temperature and requires further modelling over a temperature range between 200 and 450 K (Vereecken et Nozière, 2020). Studies show that this H-shift and autoxidation mechanism continues beyond 450 K and increases to 600K depending on the chemical nature of the compound (Belhadj et al., 2021). Among the numerous studies available in the literature on the oxidation of terpenes, limonene has the particularity of having endocyclic and exocyclic bonds which favor the formation of SOA.

Whereas in atmospheric chemistry peroxy radicals self- and cross-reactions are very important (Wallington et al., 1992), in combustion (Bailey et Norrish, 1952;Benson et al., 1981;Cox and Cole, 1985;Morley et al., 1987), it is commonly accepted that the low-temperature oxidation of hydrocarbons (RH), also named cool-flame, can lead to the formation of oxygenated intermediates, but generally, it is assumed that the autoxidation proceeds through the formation of keto-hydroperoxides (KHPs) which provide chain branching by decomposition: RH + OH $\leftrightarrows$ R + $H_2O$, R+ $O_2$ $\leftrightarrows$ ROO, ROO $\leftrightarrows$ QOOH, QOOH + $O_2$ $\leftrightarrows$

OOQOOH, OOQOOH $\leftrightarrows$ HOOQ'OOH $\leftrightarrows$ HOOQ'O + OH, HOOQ'O $\leftrightarrows$ OQ'O + OH. However, recent studies reported the formation of HOMs during the so-called low-temperature oxidation (500–600 K) of hydrocarbons and other organics, e.g., alcohols, aldehydes, ethers, esters (Wang et al., 2018;Wang et al., 2017b;Belhadj et al., 2020). There, the H-atom transfer in the OOQOOH intermediate does not involve the H-C-OOH group but another H-C group, opening new oxidation pathways. Such alternative pathways do not yield ketohydroperoxides, and a third $O_2$ addition to HOOQ'OOH yielding $OOQ'(OOH)_2$

can occur. This sequence of reactions can proceed again, yielding highly oxygenated products (Wang et al., 2017b;Belhadj et al., 2020;Belhadj et al., 2021). Also, QOOH can decompose via: QOOH → OH + cyclic ether, QOOH → OH + carbonyl + olefin, and QOOH → $HO_2$ + olefin. In few studies devoted to the understanding of atmospheric oxidation mechanism of hydrocarbons yielding highly oxidized products, autoxidation was proposed as a pathway to organic aerosols, e.g. (Jokinen et al., 2014a;Jokinen et al., 2015;Mutzel et al., 2015;Berndt et al., 2016;Crounse et al., 2013;Ehn et al., 2014). The early H-shift,

ROO $\leftrightarrows$ QOOH, is favored by increased temperature, which explains its importance in autoignition, but the presence of substituents such as OH, C=O, and C=C in the ROO radical can significantly increase the rate of H-shift making it of significance at atmospheric temperatures (Bianchi et al., 2019).

Beside these processes, the Waddington mechanism (Ray et al., 1973), involving OH and $O_2$ successive additions on a C=C double bond, followed by H-atom transfer from –OH to –OO, can occur, yielding carbonyl compounds: R-C=C-R' + OH $\leftrightarrows$

R-C-C(-R')-OH, R-C-C(-R')-OH + $O_2$ $\leftrightarrows$ OO-C(-R)-C(-R')-OH $\leftrightarrows$ HOO-C(-R)-C(R')-O $\leftrightarrows$ OH + R-C=O + R'-C=O. The Korcek mechanism (Jensen et al., 1981) through which γ-ketohydroperoxides are transformed into a carboxylic acid and a carbonyl compound can occur too. The formation of carboxylic acids and carbonyl products via the Korcek mechanism has already been postulated by Mutzel et al. (Mutzel et al., 2015) whereas it is frequently considered in recent kinetic combustion modeling (Ranzi et al., 2015).

Then, questions arise: what are the similarities and differences between the products of oxidation of hydrocarbons under simulated atmospheric conditions and cool-flames? Do oxidation routes observed in autoxidation (cool flames) play a significant role under atmospheric conditions? Can atmospheric chemistry benefit from combustion chemistry studies and vice versa?

The aim of this work is to characterize the oxidation products of limonene and more particularly those resulting from chain branching. The identification of the isomers resulting from the oxidation of $RO_2$ will be carried out thanks to a UHPLC-Orbitrap coupling in tandem mode. The initial oxidation concentrations of hydrocarbons used in the laboratory, of the order of a few ppm, only allow the detection of the presence of HOMs without being able to exploit their fragmentation and their chemical speciation within a mixture of isomers. The confirmation of the reaction mechanisms remains difficult.

To overcome this limitation, and to form these compounds in gas-phase oxidation processes with a concentration compatible with the UHPLC separation, the ionization mode, the transfer function, the fragmentation and low residence time, we chose to increase the initial concentration of limonene. Given these conditions, the study focused only on the mechanistic reaction and qualitative aspects of chemical speciation.

The impact of this initial concentration and the experimental conditions on the range of chemical formulae was investigated
by comparing our results to other limonene oxidation studies chosen for their experimental diversity (i.e. oxidation mode, concentration, type of characterization, aging time)

To this end, we studied the oxidation of limonene-oxygen-nitrogen mixtures in a jet-stirred reactor (JSR) at atmospheric pressure, large excess of oxygen, and elevated temperature. Our results are compared to literature data obtained under tropospheric relevant conditions where terpenes are oxidized by OH and/or ozone. Inventory and chemical speciation of
oxidation products, as well as the comparison with products of other modes of oxidation (ozonolysis, OH$^\bullet$ and photolysis) should lead to a better comprehension of the specificities of each oxidation mode and provide new target data for field experiments. For sake of clarity, the present oxidation experiments will be called "autoxidation" in the following sections.

## 2. Experiments

The present experiments were carried out in a fused silica jet-stirred reactor (JSR) setup presented earlier (Dagaut et al., 1986)
and used in previous studies (Dayma et al., 2011;Dagaut and Lecomte, 2003;Dagaut et al., 1998). As in earlier works (Thion et al., 2017;Dayma et al., 2011) limonene (R)-(+) (>97% pure from Sigma-Aldrich) was pumped by an HPLC pump (Shimadzu LC10 AD VP) with an online degasser (Shimadzu DGU-20 A3) and sent to a vaporizer assembly where it was diluted by a nitrogen flow. Limonene and oxygen were sent separately to a 42 mL JSR to avoid oxidation before reaching the injectors (4 nozzles of 1 mm I.D.) providing stirring. The flow rates of nitrogen and oxygen were controlled by mass flow meters. Good
thermal homogeneity along the vertical axis of the JSR was recorded (gradients of < 1 K/cm) by thermocouple measurements (0.1 mm Pt-Pt/Rh-10% wires located inside a thin-wall silica tube). In order to be able to observe the oxidation of limonene,

which is not prompt to strong autoignition (cetane number of 20, similar to that of iso-octane), the oxidation of 1% limonene ($C_{10}H_{16}$) under fuel lean conditions (equivalence ratio of 0.25, 56 %$O_2$, 43 %$N_2$) was performed at 590 K, atmospheric pressure, and at a residence time of 2 s. Under these conditions, the oxidation of limonene is initiated by H-atom abstraction by molecular
oxygen. The radicals of the fuel rapidly react with $O_2$ to form peroxy radicals which undergo further oxidation, as presented in the introduction. The absence of ozone, and no need for the addition of a scavenger, allows probing reaction mechanisms and observing chemical species potentially specific to the oxidation by OH radical.

A low-pressure sonic probe was used to freeze the reactions and take samples. To measure low-temperature oxidation products
ranging from hydroperoxides, ketohydroperoxides (KHPs), to highly oxidized molecules, the sonic probe samples were bubbled into cooled acetonitrile (UHPLC grade ≥99.9, T= 0°C, 250 mL) for 90 min. The resulting solution was stored in a freezer at -30°C. Analyses were performed by direct sample instillation (flow injection analyses heated electrospray ionization/atmospheric pressure chemical ionization :FIA HESI/APCI) settings sheath gas 12 arbitrary units (a.u.). auxiliary gas flow 0, vaporizer temperature 120°C, capillary temperature 350°C, spray voltage 3.8 kV, flow injection of 3μL/min
recorded for 1 min for data averaging) in the ionization chamber of a high resolution mass spectrometer (Orbitrap® Q-Exactive from Thermo Scientific, mass resolution of 140,000 and mass accuracy <0.5 ppm RMS). Mass calibrations in positive and negative HESI were performed using Pierce[TM] calibration mixtures (Thermo Scientific). Ultra-high pressure liquid chromatography (UHPLC) analyses were performed using an analytical column at a controlled temperature of 40°C ($C_{18}$ Phenomenex Luna, 1.6μm, 100 Å, 100x2.1 mm) for products separation after injection of 3 μL of sample eluted by water-
acetonitrile (ACN) at a flow rate of 250 μL/min (gradient 5% to 90% ACN, during 14 min). Both heated electrospray ionization (HESI) and atmospheric chemical ionization (APCI) were used in positive and negative modes for the ionization of products. APCI settings were: spray voltage 3.8 kV, vaporizer temperature of 120°C, capillary temperature of 300°C, sheath gas flow of 55 a.u., auxiliary gas flow of 6 a.u., sweep gas flow of 0 a.u., corona current of 3μA. In HESI mode, setting were : spray voltage 3.8 kV, T vaporizer of 120°C, T capillary 300°C, sheath gas flow of 12 a.u., auxiliary gas flow of 6 a.u., sweep gas
flow of 0 a.u., mass range between 50 and 1000 Da. Because oxidation of analytes in HESI has been reported previously (Pasilis et al., 2008;Chen and Cook, 2007), we verified that no significant oxidation occurred in the HESI and APCI ion sources by injecting a limonene-ACN mixture.  The optimization of the Orbitrap ionization parameters in HESI and APCI did not show any clustering phenomenon on pure limonene. The parameters evaluated were: injection source - capillary distance, vaporization and capillary temperatures, potential difference, injected volume, flow rate of nitrogen in the ionization source).
The APCI source, more versatile on polarities and adapted to low masses, was used to identify, in a second phase, KHPs and HOMs. This source, used in positive mode (minimization of salt adducts) has improved the response of low masses necessary for fragmentation. Nevertheless, it should be considered that some of the molecules presented in this study could result from our experimental conditions (continuous-flow tank reactor, concentration of reagents, temperature, reaction time) and to some extent to our acquisition conditions, different from those in the cited studies (Table 1). Indeed, the use of a continuous-flow

tank reactor operating at elevated temperature , as well as a high initial concentration of reactants can induce the formation of unrealistic atmospheric compounds. With regards to  the MS acquisition parameters, the selected mass scan range has an influence on the ion transmission, especially at the higher mass range. Figure 9s, in the supplementary material, compares two spectra of oxidized limonene with different acquisition mass ranges. A decrease in trapping efficiency at higher masses is clearly visible when changing the mass scan range from $m/z$ 150-750 to 50-750. It is also necessary to consider the possible

formation of non-covalent artifacts, without excluding an incidence on the DBE number. A more detailed description of these technical aspects is available in a recent review (Hecht et al., 2019).

To determine the structure of limonene oxidation products, MS-MS analyses were performed at collision cell energy of 10–30 eV. 2,4-Dinitrophenylhydrazine (DNPH) was also used to characterize carbonyl compounds. As in previous work (Wang et al., 2017b;Belhadj et al., 2020), the fast OH/OD exchange was used to prove the presence of hydroxyl or hydroperoxyl

functional groups in the products. We added 300 µL of $D_2O$ (Sigma-Aldrich) to 1.5 mL of sample. The resulting solution was analyzed by flow injection and HESI/APCI mass spectrometry.

## 3. Data Processing

High-resolution mass spectrometry (HR-MS) generates a significant amount of data that is easier to interpret with two- or three-dimensional visualization tools (Nozière et al., 2015;Wang et al., 2017a;Walser et al., 2008;Tu et al., 2016). In this study,

we used Kendrick's mass analysis, double bond equivalent (DBE), van Krevelen diagrams, and carbon oxidation state ($OS_c$). Kendrick's mass analysis (Sleno, 2012;Hughey et al., 2001;Kune et al., 2019;Kendrick, 1963) allows representing in two dimensions and in a new reference frame, a complex mass spectrum of an organic mixture. This reference frame is based on a mass defect calculated from structural units ($CH_2$, O, CHO). In a Kendrick representation, the homologous series (constructed by the repeated addition of structural units $CH_2$, O, CHO) are aligned on the same horizontal line. This mass defect is calculated

by the difference between the Kendrick mass and the nominal mass. In this study, $CH_2$ was chosen as the structural unit.

In Kendrick's plots, the X-axis represents the Kendrick Mass

$$(CH_2) = observed\ mass * \frac{nominal\ mass\ of\ CH_2}{exact\ mass\ of\ CH_2},$$

and the Y-axis represents the Kendrick Mass Defect

$$(CH_2) = nominal\ mass - Kendrick\ mass\ (CH_2)$$

The belonging of unknown chemical compounds to an homogeneous series of compounds can be used for their identification. The number of double bond equivalent (DBE) represents the sum of the number of unsaturation and ring present in a compound (Nozière et al., 2015). The decrease in the number of hydrogen atoms increases its value, but it is independent of the number

of oxygen atoms. It can be used to identify certain groups of chemical compounds or reaction mechanisms (Kundu et al., 2012).

The decimal values of this number were not taken into account in this study. The van Krevelen diagram (Kim et al., 2003;Van Krevelen, 1950) shows the evolution of the H/C ratio as a function of O/C for a set of identified molecules. In a complex organic mixture, it allows classifying the chemical products according to their degree of oxidation or their degree of reduction/saturation. This type of representation allows the identification of classes of compounds such as aliphatics, aromatics, or highly oxidized compounds (Fig. 3).

The oxidation state of carbon allows the degree of oxidation of organic species (alcohols, aldehydes, carboxylic acids, esters, ethers, and ketones, but not peroxides) (Kroll et al., 2011) to be measured. It is defined by the simple equation:

$$OSc \approx 2O/C - H/C$$

This data can be used together with the atomic ratios of van Krevelen's diagrams to identify families of organic compounds (Tu et al., 2016;Wang et al., 2017a;Bianchi et al., 2019). In the case of HOMs, three families of compounds can be distinguished according to their oxidation state and the O/C and H/C ratios:

$O/C \geq 0.6$ and $OS_c \geq 0$ (Region 1, highly oxygenated and highly oxidized)

$O/C \geq 0.6$ and $OS_c < 0$ (Region 2, very oxygenated and moderately oxidized)

$OS_c \geq 0$ and $H/C \leq 1.2$ (Region 3, moderately oxygenated and highly oxidized)

## 4. Results and discussion

The oxidation of 1% limonene ($C_{10}H_{16}$) was studied at 590 K, atmospheric pressure, and at a residence time of 2 s. Under these conditions, the fuel conversion is moderate but formation of low-temperature oxidation products is maximized.

To study the nature of the chemical products formed and the particularity of autoxidation, we compared our results with those obtained by ozonolysis and OH-initiated photooxidation of limonene. This comparison was carried out using visualization methods adapted to large intrinsic data sets of high resolution and high sensitivity reached with current mass spectrometry. At this scale, these tools allow differentiating families of compounds or chemical processes that are hardly perceptible at the level of a few individuals (chemical species).

The comparison of the oxidation modes of limonene (autoxidation and ozonolysis/photooxidation) was based only on the nature of the chemical formula of products, without considering the quantitative, sensitivity, or ionization aspects that are difficult to exploit given the diversity of chemical products formed in this study (i) of the analytical methods, and (ii) the large number of instruments involved in this comparison.

To carry out this comparison, and in order to obtain the greatest representativeness of the oxidation of limonene by ozonolysis and OH-initiated photooxidation, we have selected nine previous studies for their diversity of oxidation and characterization processes. Table 1 presents the main experimental parameters of these studies.

**Table 1.** Main experimental parameters for studies of limonene oxidation.

| Ref | Oxidation | Sampling | Experimental Setup | Initial concentrations of reactants | Ionization source | Instrument |
|---|---|---|---|---|---|---|
| (Fang et al., 2017) | OH-initiated photooxidation dark ozonolysis | online | smog chamber | 500 ppb of ozone 900–1500 ppb of limonene | UV; 10 eV | Time-of-Flight (ToF) |
| (Witkowski and Gierczak, 2017) | Dark ozonolysis | off-line | flow reactor | 0.15 to 4.0 ppm ozone Limonene concentration not provided | ESI; 4.5 kV | Triple quadrupole |
| (Jokinen et al., 2015) | Ozonolysis | online | flow glass tube | 6.1-6.9 $\times 10^{11}$ molec.cm$^{-3}$ of ozone 1–10000 $\times 10^{9}$ molec.cm$^{-3}$ of limonene | chemical ionization | Time-of-Flight (ToF) |
| (Nørgaard et al., 2013) | Ozone (plasma) | online | Direct on the support | 850 ppb ozone 15-150 ppb limonene | plasma | Quadrupole time-of-flight (QToF) |
| (Bateman et al., 2009) | Dark and UV radiations ozonolysis | off-line | Teflon FEP reaction chamber | 1 ppm ozone 1 ppm limonene | ESI; (not specified) | LTQ-Orbitrap Hybrid Mass Spectrometer |
| (Walser et al., 2008) | Dark ozonolysis | off-line | Teflon FEP reaction chamber | 1-10 ppm ozone 10 ppm limonene | ESI; 4.5 kV | LTQ-Orbitrap Hybrid Mass Spectrometer |
| (Warscheid and Hoffmann, 2001) | Ozonolysis | online | Smog chamber | 300-500 ppb ozone and limonene | APCI; 3kV | Quadrupole ion trap mass spectrometer |
| (Hammes et al., 2019) | Dark ozonolysis | online | Flow reactor | 400-5000 ppb ozone 15, 40, 150 ppb of limonene | [210] Po alpha | HR-ToF-CIMS |
| (Kundu et al., 2012) | Dark ozonolysis | off-line | Teflon reaction chamber | 250 ppb ozone 500 ppb limonene | ESI; 3.7 and 4 kV | LTQ FT Ultra, Thermo Scientific |
| This work | Cool-flame autoxidation | off-line | Jet-stirred reactor | 1% limonene 56 %O$_2$, 43 %N$_2$ | APCI; 3kV HESI; 3.8 kV | Orbitrap® Q-Exactive |

These nine experimental studies performed under diverse initial conditions, as shown in Table 1, yielded a first set of 1233 molecular formulae for an inventory which, although incomplete, gives a broad representativeness of the chemical products which can result from limonene ozonolysis and OH-initiated photooxidation.The second set was obtained using the chemical formulae observed here during limonene autoxidation. It should be noted that this second set of data is generated from different

experimental conditions than the previous one, i.e. with a continuous flow reactor, a high concentration of reactants, a high temperature and a reaction time of 2s. As previously mentioned, MS detector type and MS acquisition parameters (e.g. mass scan range) in the present study and those compared in the Table 1 can affect the inventory of this second set.


For the identification phase, we favored direct injection for its sensitivity. We used UHPLC for its capacity of isomers separation and advanced isomers identification through MS-MS analysis. For direct injection, we have chosen a HESI source operating in negative mode for its wide range of polar molecule ionization. After verification of the absence of oxidation induced by the ionization source (Pasilis et al., 2008;Chen and Cook, 2007) and elimination of the ions common to the reference, attribution rules were made on the basis of molecules composed solely of carbon, hydrogen and oxygen, respecting a deviation of less than 3 ppm by mass over the range 50-1000 Da. Chemical formulae with relative intensity less than 1 ppm to the highest mass peak in the mass spectrum were not considered. Following these rules, we identified 1138 chemical formulae in our oxidized limonene sample, which we split into three groups centred on the number of monomers, i.e. a limonene molecule with different degrees of oxidation: I ($50<m/z<300$); II ($300<m/z<500$) and III ($500<m/z<700$) (Kundu et al., 2012;Leungsakul et al., 2005;Nørgaard et al., 2013).



These groups are identified in Figure 1 which shows the mass spectrum of oxidized limonene. Generally speaking, these groups are built around one or more monomers and a few families of chemical reactions. Group I corresponds to compounds resulting from multiple oxidation reactions including fragmentation and condensation. Groups II and III correspond to higher molecular masses, resulting from addition and condensation reactions including, in the case of ozonolysis reactions of (i) hemiacetalization, (ii) with radicals (hydroperoxy or Griegee), and (iii) of condensation of aldols and/or esterification (Kundu et al., 2012).


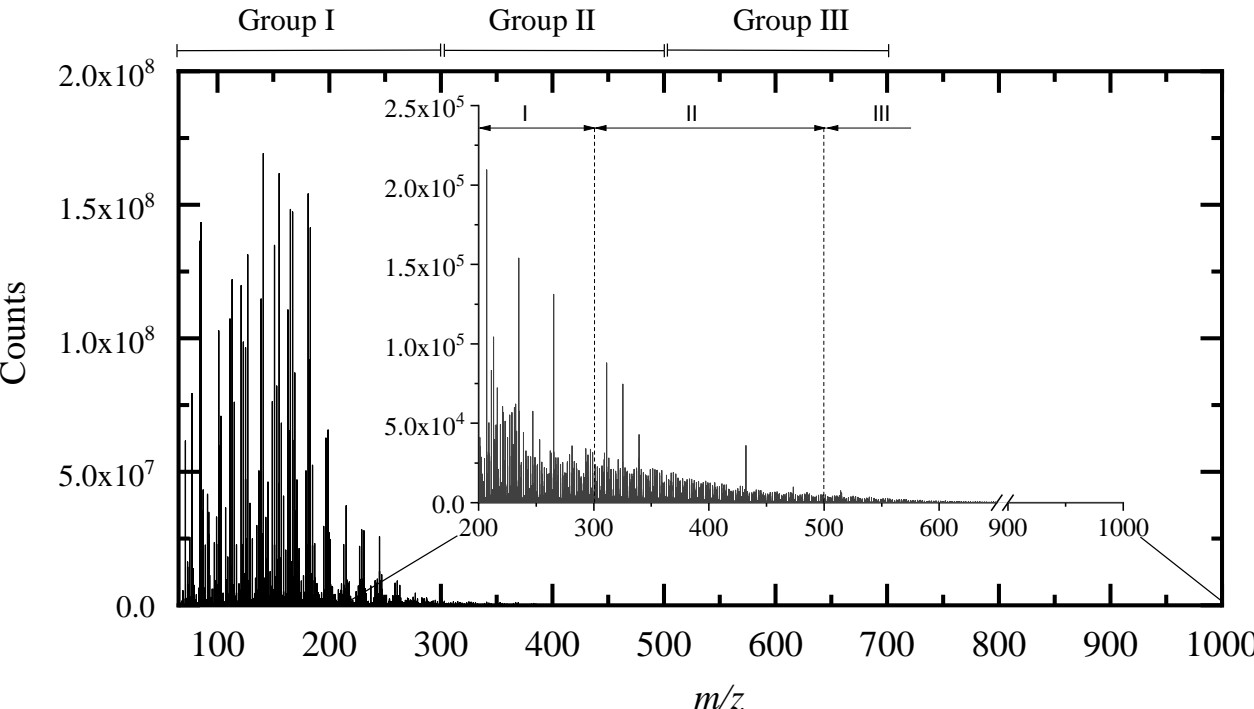

**Figure 1.** Limonene oxidation sample analyzed by FIA and HESI (negative mode, sheath gas flow of 12 a.u., auxiliary gas flow of 6 a.u., sweep gas flow of 0 a.u., voltage 3.8kV, capillary temperature 300°C, vaporizer temperature 120°C, Hamilton syringe at a flow rate of 3 μL min−1).

The two sets obtained from each oxidation mode were merged, forming a new set of 1600 molecules. In this set, ~50% of the chemical formulae (771) are common to both oxidation modes, while 462 molecular formulae are obtained solely by ozonolysis/ photooxidation and 367 are produced in autoxidation experiments only.

All the molecular formulae were represented in a Kendrick diagram (based on a $CH_2$ structural unit) associated with the DBE number (Fig. 2). The representation of Kendrick highlights the families of compounds, the addition of a third dimension makes it possible to study the reaction mechanisms linking these different families (Bateman et al., 2009); (Kundu et al., 2012).

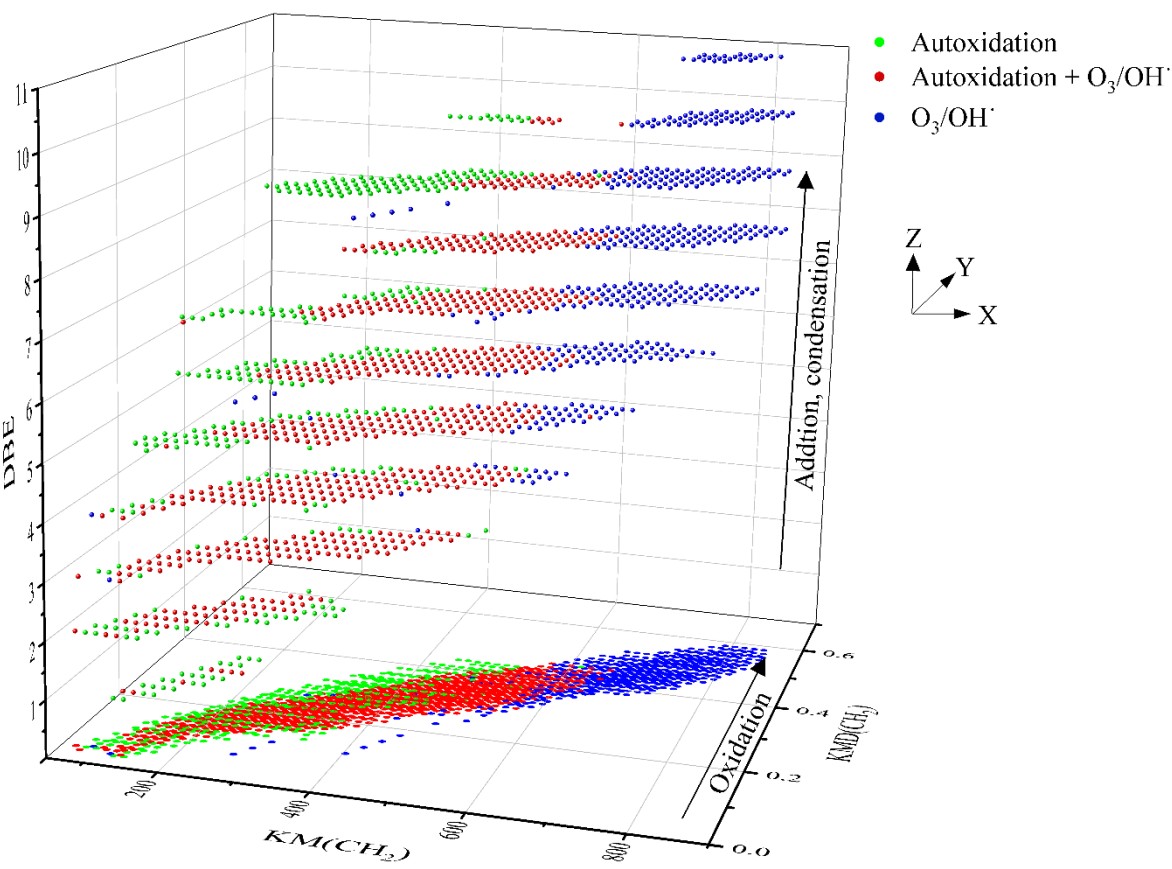

**Figure 2.** All the chemical products, resulting from limonene oxidation by ozonolysis/photooxidation and autoxidation gathered in the form of a Kendrick diagram correlated to the DBE (with projection on the XY plane): ● new chemical products from autoxidation experiments (JSR) ● common to the 3 modes of oxidation ● chemical products with molecular formulae not observed in autoxidation.

The chemical formulae are distributed into eleven XY planes according to their DBE number. The representation of each of these XY plans is given in the S1 supplementary material. This representation immediately highlights the origin of the chemical formulae, i.e. those specific to autoxidation, common to the three modes of oxidation, or specific to ozonolysis and photooxidation. The chemistry of these chemical species can then be read graphically on the axes: the X-axis represents the extent of the family, the Y-axis represents oxidation and the Z-axis denotes addition and condensation reactions. This is illustrated in Figure S1 of the Supplementary material.

Unexpectedly, the distribution of chemical formulae is homogeneous and forms a continuum between oxidation by autoxidation and by ozonolysis/photooxidation.

In autoxidation, the above mentioned pathways, probably constrained by short oxidation time (residence time of 2 s), seem to extend the molecular weight growth of products up to ~300 Da, favor splitting (decrease of DBE), additions or condensation (increase of DBE) of unsaturated chemical groups. Ozonolysis and photooxidation experiments, performed over longer periods of time (several seconds < t < few hours), promote, in addition to the previous reactions, the appearance of oligomers and an increase of products molecular weight (Zhao et al., 2015).

In the Kendrick diagram (Fig.2), the DBE=3 XY plane containing limonene includes chemical formulae of products deriving from the early oxidation steps. In this case, the breaking of unsaturation, double bond, and ring are counterbalanced by the formation of carbonyls or ozonides, keeping DBE constant. In this XY plane, 83% of the chemical formulae of products obtained by autoxidation are identical to those formed by ozonolysis/photooxidation, although ozone is most likely absent from JSR autoxidation experiments. Many of the molecular formulae of products described in previous ozonolysis/photooxidation works involve reaction mechanisms based on ozonides and Criegee intermediates. These molecular formulae can be primary products, i.e. weakly oxidized (Bateman et al., 2009) ($C_9H_{14}O_3$, $C_9H_{14}O_4$, $C_{10}H_{14}O_3$, $C_{10}H_{14}O_4$, $C_{10}H_{16}O_2$, $C_{10}H_{16}O_3$, $C_{10}H_{16}O_4$) but also secondary, with an increasing number of oxygen atoms ($C_7H_{10}O_4$, $C_7H_{10}O_5$, $C_8H_{12}O_3$, $C_8H_{12}O_4$, $C_8H_{12}O_5$, $C_9H_{14}O_3$, $C_9H_{14}O_4$, $C_9H_{14}O_5$, $C_{10}H_{16}O_3$) (Hammes et al., 2019) and ($C_7H_{10}O_6$, $C_8H_{12}O_6$, $C_9H_{14}O_6$) (Kundu et al., 2012). Additional chemical formulae observed in the current study (also referred as 'new chemical formulae' below) do not form a new group, but are in the continuity of the families of chemical molecules found in previous studies (Witkowski and Gierczak, 2017;Jokinen et al., 2015;Walser et al., 2008;Kundu et al., 2012;Fang et al., 2017;Nørgaard et al., 2013;Bateman et al., 2009;Warscheid and Hoffmann, 2001;Hammes et al., 2019). These families are built on the basis of a simple difference in alkyl groups ($CH_2$ basic unit of the Kendrick diagram).

The decrease of DBE from 3 to 2 and then to 1 reflects a greater reactivity on double bonds and on the limonene ring. This reactivity on unsaturated sites is accompanied by a fragmentation of the C-C bonds and a decrease in molecular weight. The new molecular formulae of products, specific to autoxidation, are located at the extremities of the DBE=2 plane or on almost the entire DBE=1 plane. They are characterized by a lower O/C ratio, which can be explained by less advanced oxidation, and certainly by fragmentation. The observation of these new chemical formulae of products, compared to previous studies, can be explained, in addition to the short oxidation time and the elevated temperature in the JSR experiment, by a termination in the radical chain process or by bimolecular reactions (Rissanen et al., 2014;Walser et al., 2008).

The increase in DBE from 3 to 11 characterizes the increase in double bonds and degree of unsaturation obtained by the addition or condensation of limonene oxidized species. In the case of ozonolysis/photooxidation, this increase is usually explained by reactions between Criegee radicals (Criegee intermediate) and acids or alcohols, by hemiacetal formation, aldolization, or esterification (Bateman et al., 2009;Docherty et al., 2005). Each of these reactions is associated with an increase in DBE (hemiacetal: +2; aldolization: +3). For example, in the case of aldol condensation between two aldehydes

(limonoaldehyde + 7-OH-limonoaldehyde), the DBE increases from 3 to 6. The emergence of new chemical formulae of

products with DBE=9 is mostly observed for molecular weights between 200 and 500 Da. These molecules probably

correspond to the addition or condensation of several oxidized limonene compounds (condensation/addition of several cycles)

favored, under short time oxidation, by the elevated experimental temperature (590 K).

In general, the new molecular formulae of products observed in autoxidation, compared to ozonolysis (Tu et al., 2016), have

a lower molecular weight and are better found in group I ($m/z < 300$, 52%) associated with addition or splitting reactions

around the limonene skeleton (monomer channel).

In ozonolysis/photooxidation, the products which molecular formulae are rather located in groups II and III can easily

oligomerize. This process is highly time-dependent in the presence of ozone (Kundu et al., 2012).

In addition to the identification of chemical families by Kendrick's analysis, it is possible to specify the nature of the chemical

products using a van Krevelen diagram. Figure 3 shows a representation of all the chemical formulae of products observed in

this work, overlayed on the locations of the different families of chemical compounds described in the literature (Bianco et al.,

2018;Nozière et al., 2015). These families are defined by O/C and H/C ratios and shown in Figure 3.

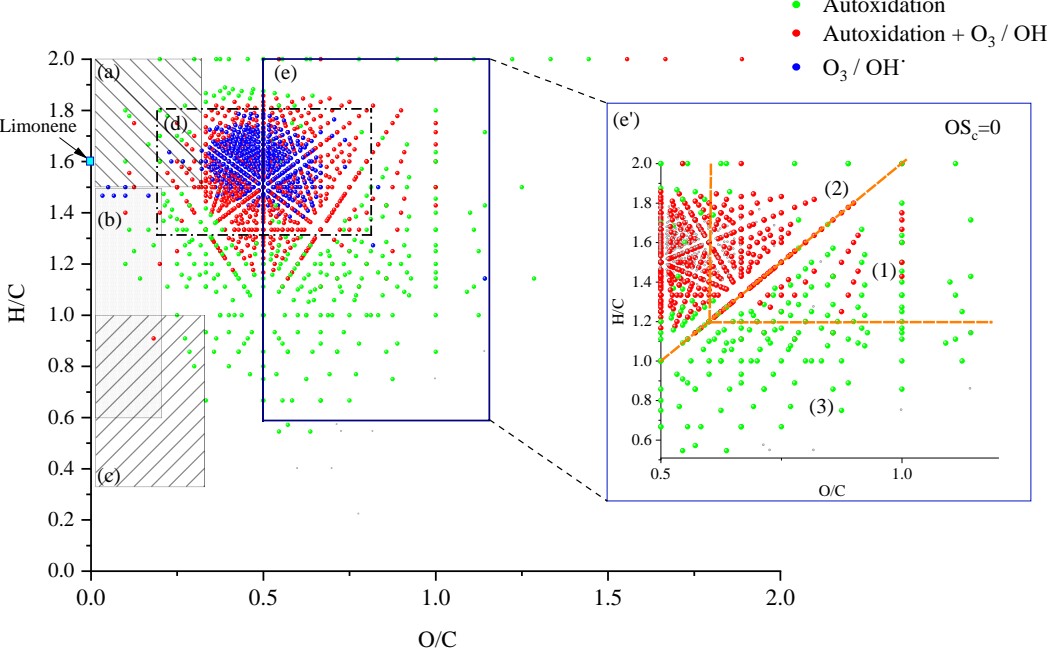

**Figure 3.** van Krevelen representation of all the chemical products and the different families of chemical compounds described

in the literature: (a) aliphatic compounds; (b) unsaturated hydrocarbons; (c)aromatics hydrocarbons ; (d) compounds obtained

by ozonolysis (including OH-initiated oxidation) (Kundu et al., 2012); (e) HOMs, (e') HOMs formed without $O_3/OH^{\cdot}$.

In the limonene oxidation process, observable from the left to the right on this figure, the first oxidation steps concern both the products resulting from JSR autoxidation and those from ozonolysis/photooxidation.

## 4.1 Characterization of KHPs

Oxidation, at the initial stage, forms compounds with number of carbon atoms varying from 7 to 11, number of hydrogen atoms ranging from 12 to 16, and up to 9 oxygen atoms. These chemical compounds are mainly in the (d) area. In ozonolysis/photooxidation studies, initial oxidation phase, by ozone and/or the radical OH˙ have been widely described (Walser et al., 2008;Kundu et al., 2012;Nørgaard et al., 2013;Librando and Tringali, 2005). Although the experimental conditions and the associated reaction mechanisms are different, it is observed, similarly to Fang et al (Fang et al., 2017), many

common products of these different oxidation modes. In the case of autoxidation, in the absence of ozone, H-abstraction by the radical OH˙ initiates further oxidation steps yielding ROO˙ radicals and hydroperoxides.

$$RH + OH \leftrightarrows R˙ + H_2O \; ; \; R˙ + O_2 \leftrightarrows ROO˙ \; ; \; ROO˙ + R´H \leftrightarrows ROOH + R´; ROO˙ + HOO˙ \rightarrow ROOH + O_2$$

These reactions will themselves lead to the formation of, among others, KHPs, diketones, or will proceed further (see Section 1) and lead to the formation of HOMs (Jokinen et al., 2014b;Wang et al., 2019;Wang et al., 2016). In the initial stage of this autoxidation, we studied the formation of the compounds $C_{10}H_{14}O_3$, $C_{10}H_{12}O_2$, $C_{10}H_{16}O_2$, $C_{10}H_{14}O_{5-11}$ corresponding respectively to the chemical formulae of KHPs, diketones, oxyhydroxides, and HOMs. Experiments were carried out using UHPLC-Orbitrap coupling in tandem mode in order to isolate these molecules and fragment them with an HCD at energy

ranging from 10 to 30 eV. Considering a mass range of initial molecules detected in autoxidation lower than 700 Da, we used an APCI source in positive mode, well suited to this mass range.

For the KHPs ($C_{10}H_{14}O_3$), whose mechanisms of formation from limonene are described in the Supplementary material S4 (with example of reaction mechanisms of KHP formation and probable sites of attack of OH radicals), the analyses by high resolution mass spectrometry confirm the presence of 12 isomers among the compounds formed, compared to a maximum of

18 potentially produced (S5).

Among these 12 isomers, MS/MS fragmentation allowed the identification of three groups of compounds presented in Figure 4. The limits of separation and detection of these chemical compounds made it impossible to specify the position of the functional groups in these isomers.

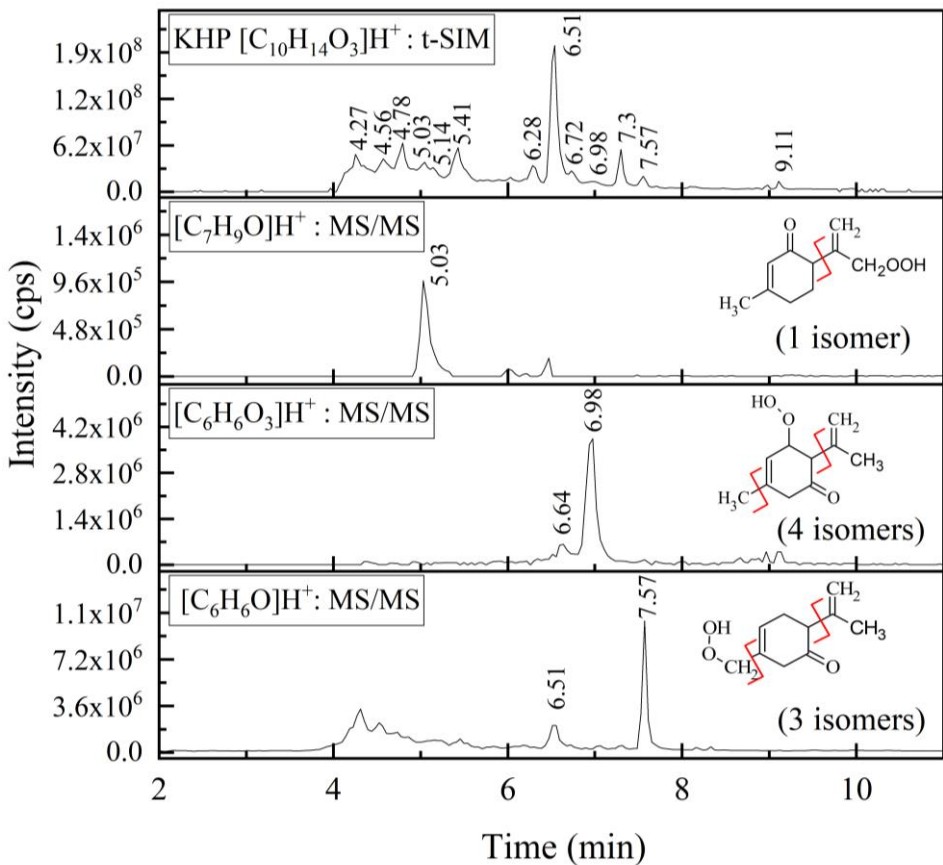

**Figure 4.** MS analyses of the KHPs isomers together with their fragmentation and number of possible isomers for each group (S5).

In order to verify the presence of carbonyl groups, 20 μl of a mixture containing DNPH (20 μl of $H_3PO_4$ (85%) in ACN with 20 μl of 2,4-DNPH) were added to 1 ml of sample. This mixture was allowed to react for 4 hours before analysis. Characterization, at different reaction times (0.5, 1, and 4 hours), was performed by UHPLC-MS APCI (⁻) in tSIM mode following 361.1153 Da mass of the $C_{16}H_{18}O_6N_4$ compound (Fig. S6). One could note an increase in the intensity of the signal (inset Fig. S6). Nearly 12 isomers were observed, with an elution time that is longer and consistent with the initial retention time of the KHPs, thus confirming the presence of carbonyl groups. However, the fragmentation carried out on all these chromatographic peaks did not make it possible to complete the chemical speciation of all the isomers. Nevertheless, for the first time, this study confirms the formation of KHPs initiated by the OH˙ radical during the oxidation of limonene.

The difficulty for characterizing KHPs lies in the fact that these products are unstable and transform according to different mechanisms. One of the instabilities described in the literature consists in spontaneous dehydration of KHPs to give diketones

(Herbinet et al., 2012). Analysis of the data confirms the presence of a diketone ($C_{10}H_{12}O_2$). It shows that diketones are detected both by elution of oxidized limonene (7.79 and 8 min), and systematically in the chromatogram of KHP isomers. This means that this spontaneous transformation can occur in the spectrometer without questioning the presence of diketones in the initial sample. Figure S7 in the Supplementary material compares the two profiles of diketones resulting from the fragmentation of KHPs and by elution.

Other transformation pathways of KHPs are possible, e.g., via the Korcek mechanism (Jensen et al., 1981, Mutzel et al., 2015) where δ-KHPs decompose into carbonyl compounds and carboxylic acids. Among the 18 proposed KHPs (Supplementary material S5), 4 isomers (#4, 11, 15, and 18) could react via the Korcek mechanism, but only the #4 isomer is likely to form a cyclic peroxide between a carbonyl group and a hydroperoxide group. The other three isomers will give, after ring opening, isomers of the compound $C_{10}H_{14}O_3$. For the #4 isomer, the Korcek mechanism leads to the formation of a carbonyl compound, $C_9H_{12}O$, and the formic acid $CH_2O_2$ (Fig. 5). UHPLC analyses confirm the presence of products with chemical formula $C_9H_{12}O$ in the form of two isomers (Fig. 5), but only the peak located at 5.64 minutes shows a $C_8H_{12}$ fragment consistent with the transformation of the initial KHP.

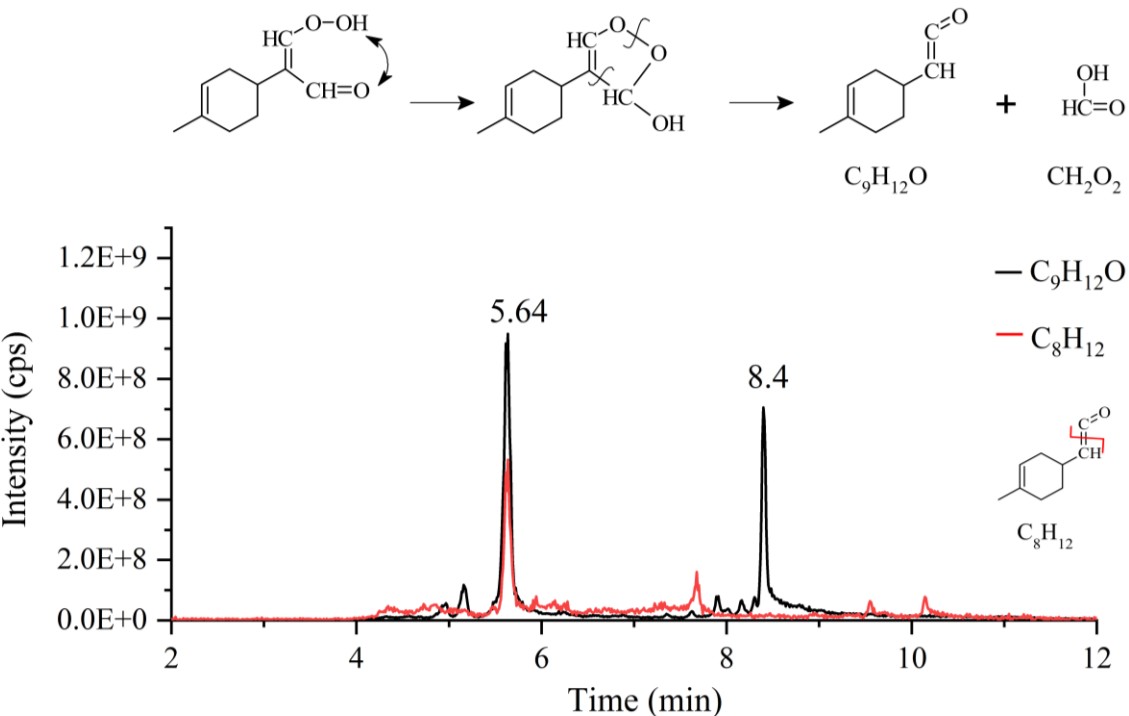

**Figure 5.** Korcek mechanism for the #4 KHP isomer and chromatograms of $C_9H_{12}O$ and the fragment $C_8H_{12}$ (APCI(+), vaporizer temperature =120°c, sheath gas flow of 50 a.u., auxiliary gas flow of 0 a.u.; sweep gas flow of 0 a.u., capillary temperature of 300°C, corona current of 3 μA).

KHPs can also give rise to branching reactions that will generate radicals promoting autoxidation by $OH^{\cdot}$ (Wang et al., 2016).

$$R-\underset{\underset{O}{\|}}{C}-CH_2-\underset{\underset{\underset{OH}{|}}{O}}{CH}-R' \quad \longrightarrow \quad R-\underset{\underset{O}{\|}}{C}-CH_2-\underset{\underset{O^{\cdot}}{|}}{CH}-R' \quad + \quad OH^{\cdot}$$

Otherwise, if the H-atom transfer in the OOQOOH intermediate does not involve the H-C-OOH group but another H-C group, then no ketohydroperoxide is formed and a third $O_2$ addition to HOOQ'OOH yielding OOQ'(OOH)$_2$ can occur. If oxidation proceeds further following this pathway, it can lead to the formation of HOMs.

Considering the presence of the radicals $OH^{\cdot}$, we also searched for chemical compounds resulting from the Waddington mechanism (Li et al., 2020). This mechanism, through which oxidation of alkenes can occur, has two reaction pathways in the

case of limonene. The first pathway leads to the formation of a $C_9H_{14}O$ ketone and formaldehyde via oxidation of the exocyclic double bond. The second, involving the endocyclic double bond of limonene, gives the compound $C_{10}H_{16}O_2$. For each pathway, we obtained three isomers on the chromatograms (Fig. 6).

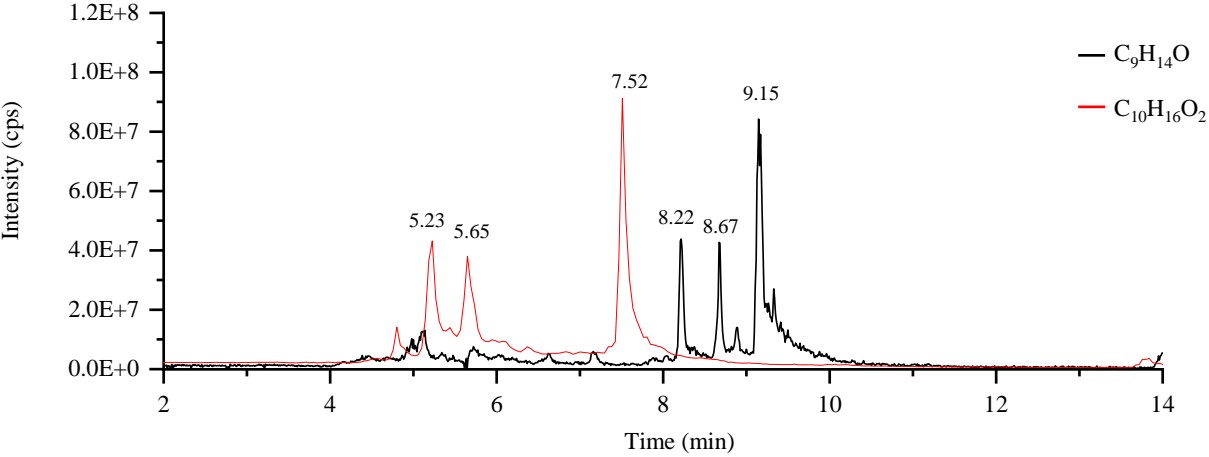

**Figure 6.** Oxidation of limonene according to the Waddington mechanism and chromatograms of the compounds obtained: $C_9H_{14}O$ and $C_{10}H_{16}O_2$ (APCI$^+$).

The fragmentation of these isomers did not allow their identification. However, in order to confirm the presence of carbonyl groups, we added 20 µl of a mixture containing DNPH (Same experimental condition as above) to 1 ml of sample, and let this mixture react for 4 hours before analyses. In order to facilitate the detection of isomers and their addition compounds, we used the APCI(-) mode for UHPLC analyses. Only the compound $C_{16}H_{20}O_5N_4$ ($C_{10}H_{16}O_2$ + DNPH) was detected. The second carbonyl compound (yielding $C_{22}H_{24}O_8N_8$) could not be confirmed (Supplement SI fig. 8). Moreover, the addition of $D_2O$, in order to test the presence of -OH or -OOH groups, gave no results for the two chemical compounds. Recent modelling work on Waddington mechanism (Lizardo-Huerta et al., 2016) has shown that structural parameters have an impact on the energy

barriers associated with the β-scission step. There, a decrease in the activation energy is observed when the substitution of the carbon atom carrying the peroxyl function increases. According to that study, this effect is amplified by the degree of substitution of the carbon atom carrying the hydroxy group. These results are in agreement with the preferential detection of the compound $C_{10}H_{16}O_2$ whose carbon atoms, with the peroxy groups, is the most substituted. Beyond the complete identification of the isomers, this study confirms the presence of chemical compounds which could well result from the

Waddington mechanism during limonene autoxidation in JSR.

## 4.2 Characterization of HOMs

Different strategies were considered for tracking the production of HOMs. Because of higher sensitivity and difficulty in separating isomers, FIA was preferred. Besides KHPs ($C_{10}H_{14}O_3$), the compounds $C_{10}H_{14}O_5$, $C_{10}H_{14}O_7$, $C_{10}H_{14}O_9$, $C_{10}H_{14}O_{11}$

(Fig. 3, area (d)) were detected by FIA (APCI +, vaporizer temperature =120°C, sheath gas flow of 12 a.u., auxiliary gaz flow of 0 a.u.; sweep gas flow of 0 a.u., capillary temperature of 300°C, corona discharge current of 3 μA, flow of 8 μl/min). We also detected $C_{10}H_{14}O_2$ (keto-hydroxide), $C_{10}H_{14}O_4$, $C_{10}H_{14}O_6$, $C_{10}H_{14}O_8$, and $C_{10}H_{14}O_{10}$ (figure 7a).Whereas the products with odd numbers of oxygen atoms can derive from 'combustion' oxidation pathways (figure 7b), as presented in the introduction, those with pair numbers of oxygen atoms can be formed via the classical atmospheric oxidation pathway yielding alkoxy

radicals, i.e. $2 \, RO_2^{\bullet} \rightarrow ROOOOR \rightarrow 2 \, RO^{\bullet} + O_2$, $RO_2^{\bullet} + HO_2^{\bullet} \rightarrow RO^{\bullet} + {}^{\bullet}OH + O_2$, and $RO_2^{\bullet} + NO \rightarrow RO^{\bullet} + NO_2$ followed by alkoxy H-shift (Baldwin and Golden, 1978; Atkinson and Carter, 1991) and peroxidation, $RO^{\bullet} \rightarrow {}^{\bullet}R_{-H}OH$; ${}^{\bullet}R_{-H}OH + O_2 \rightarrow {}^{\bullet}OOR'OH$. The reaction can continue with sequential H-shift and oxygen addition, yielding HOMs via up to six $O_2$ addition in the present study.

**(a)**

$$C_{10}H_{16} \xrightarrow{-H} C_{10}H_{15} \xrightarrow[(1)]{+O_2} C_{10}H_{15}O_2 \xrightarrow{RO_2} C_{10}H_{15}O_4C_{10}H_{15} \longrightarrow O_2 + 2\,C_{10}H_{15}O \xrightarrow{\text{H-shift}} C_{10}H_{14}OH$$

Limonene    $(R^{\cdot})$    $(RO_2^{\cdot})$    $(RO^{\cdot})$    $(Q^{\cdot}OH)$

$$C_{10}H_{14}OH \xrightarrow[(2)]{+O_2} C_{10}H_{15}O_3 \xrightarrow[+O_2\ (3)]{\text{H-shift}} C_{10}H_{15}O_5 \xrightarrow[+O_2\ (4)]{\text{H-shift}} C_{10}H_{15}O_7 \xrightarrow[+O_2\ (5)]{\text{H-shift}} C_{10}H_{15}O_9 \xrightarrow[+O_2\ (6)]{\text{H-shift}} C_{10}H_{15}O_{11}$$

$(Q^{\cdot}OH)$    $^{\cdot}OOQOH$    $HOOQ'(OH)OO^{\cdot}$    $(HOO)_2Q''(OH)OO^{\cdot}$    $(HOO)_3P(OH)OO^{\cdot}$    $(HOO)_4P'(OH)OO^{\cdot}$

(each with HO· / H-shift arrows downward)

$$C_{10}H_{14}O_2 \qquad C_{10}H_{14}O_4 \qquad C_{10}H_{14}O_6 \qquad C_{10}H_{14}O_8 \qquad C_{10}H_{14}O_{10}$$

$OQ'OH$    $HOOQ''(OH)O$    $(HOO)_2P(OH)O$    $(HOO)_4P'(OH)O$    $(HOO)_6P''(OH)O$

**(b)**

$$R^{\cdot} \xrightarrow[(1)]{+O_2} ROO^{\cdot} \xrightarrow{\text{H-shift}} QOOH \xrightarrow[(2)]{+O_2} {}^{\cdot}OOQOOH \xrightarrow{\text{H-shift}} HOOQ'OOH \xrightarrow[(3)]{+O_2} (HOO)_2Q'OO^{\cdot}$$

(HO· / H-shift downward)

$$HOOQ'O \qquad\qquad (HOO)_2Q''O$$

$$(HOO)_2Q'OO^{\cdot} \xrightarrow{\text{H-shift}} (HOO)_2Q''OOH \xrightarrow[(4)]{+O_2} (HOO)_3Q''OO^{\cdot} \xrightarrow{\text{H-shift}} (HOO)_3POOH \xrightarrow[(5)]{+O_2} (HOO)_4POO^{\cdot}$$

(HO· / H-shift downward)

$$(HOO)_3PO \qquad\qquad (HOO)_4P'O$$

$$(HOO)_4POO^{\cdot} \xrightarrow{\text{H-shift}} (HOO)_4P'OOH \xrightarrow[(6)]{+O_2} (HOO)_5P''OO^{\cdot} \xrightarrow[-OH]{\text{H-shift}} (HOO)_5P'''O$$

**Figure 7.** Reaction pathways to highly oxygenated products considered in atmospheric chemistry (a) and (b) (Bianchi et al., 2019). Recently extended reaction pathways in combustion (b) (Wang et al., 2017).

The intensity of ions signal decreases with increasing number of O atoms in the $C_{10}H_{14}O_{2,4,6,8,10}$ (by 5 orders of magnitude) and $C_{10}H_{14}O_{3,5,7,9,11}$ (by 6 orders of magnitude) products. Nevertheless, the diversity of reaction pathways, associated with the increasing number of chemical compounds, makes it difficult within a population of several hundred chemical compounds to identify all HOMs. Therefore, we have used again the van Krevelen diagram, which allows following the evolution of the oxidation of the first HOMs and to identify them according to definitions that seem to be consensus (Walser et al., 2008;Tu et al., 2016;Nozière et al., 2015;Wang et al., 2017a). To this end, we used the average carbon oxidation state $OS_c$ which allows distinguishing three regions according to the nature of the functional groups: Region 1 (O/C $\geq$ 0.6 and $OS_c \geq$ 0) consists of highly oxygenated and highly oxidized compounds (acids and carbonyls), Region 2 (O / C $\geq$ 0.6 and $OS_c$ <0), consists of

highly oxygenated and moderately oxidized compounds (alcohols, esters and peroxides), finally, Region 3 ($OS_c \geq 0$ and H/C $\leq 1.2$) includes compounds with a moderate level of oxygen, but strongly oxidized (Tu et al., 2016).

It can be seen from Figure 3 that autoxidation enhances the development of HOMs, compared to ozonolysis/photooxidation, and that the majority of these new products are found in Regions 1 and 3 of the inset of Figure 3. Thus, further oxidation can go on. We observed products of addition of up to 17 oxygen atoms yielding $C_{25}H_{32}O_{17}$.

### 4.3 Complementary method of screening

Further study of the oxidation of chemical compounds and their reaction mechanisms is limited by the complexity of the exponential increase of chemical reactions, chemical species, and their isomers. Nevertheless, monitoring the evolution of these complex mixtures is possible by correlating the OSc to the number of carbon atoms ($n_c$). We have reported in Figure 8 our measurements and have associated to these results the different biogenic VOCs families defined in the literature (Low-volatility oxygenated organic aerosol (LV-OOA), semi-volatile oxygenated organic aerosol (SV-OOA), hydrocarbon-like organic aerosol (HOA), and biomass burning organic aerosol (BBOA) corresponding to particulates (Kroll et al., 2011;An et al., 2019). The development of advanced oxidation, specific to autoxidation, is confirmed with an $OS_c$ close to 1. As it stands, it is difficult to make hypotheses on the evolution of these new chemical products and, in the absence of speciation, to assess their environmental impact.

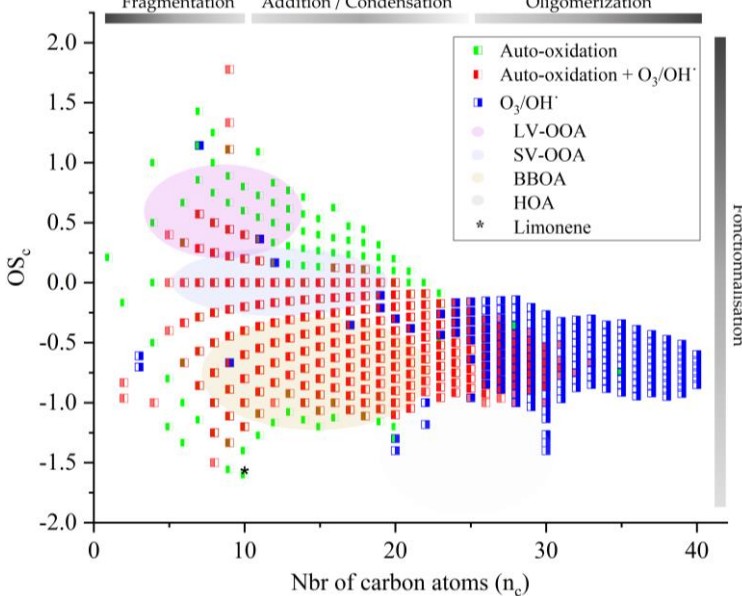

**Figure 8.** Representation in the $OS_c$-$n_c$ space of all the chemical formulae considered in this study and analysis of the degree of oxidation.

Finally, it is also possible to further exploit the van Krevelen diagram by introducing reaction mechanisms. Until now this diagram has been used to identify reaction pathways or families of compounds (Kim et al., 2003;Wu et al., 2004). One can refine this identification by associating to a reaction mechanism a vector whose amplitude and direction will allow linking reagents and products. By applying this method to all the experimental data points, one scans the space of the possibilities of formation of a compound or its isomers.

If this method is applied to the formation of $C_{10}H_{14}O_3$ (KHPs and isomers), the vector is defined by the loss of two hydrogen atoms and the gain of three oxygen atoms. By focusing only on molecules composed of 10 carbon atoms, 31 $C_{10}H_{14}O_3$ isomers were identified, 17 of which are new chemical formulae detected in autoxidation only. This is an exhaustive inventory of the possibilities of formation of these compounds based on the experimental data points and not on thermodynamic and kinetic considerations. Applying the same method to the search for keto-dihydroperoxides (-2H; +5O) and di-ketohydroperoxide (-

2H; +7O), we observed the formation of 24 and 23 compounds, respectively. All these results are presented in Figure 9.

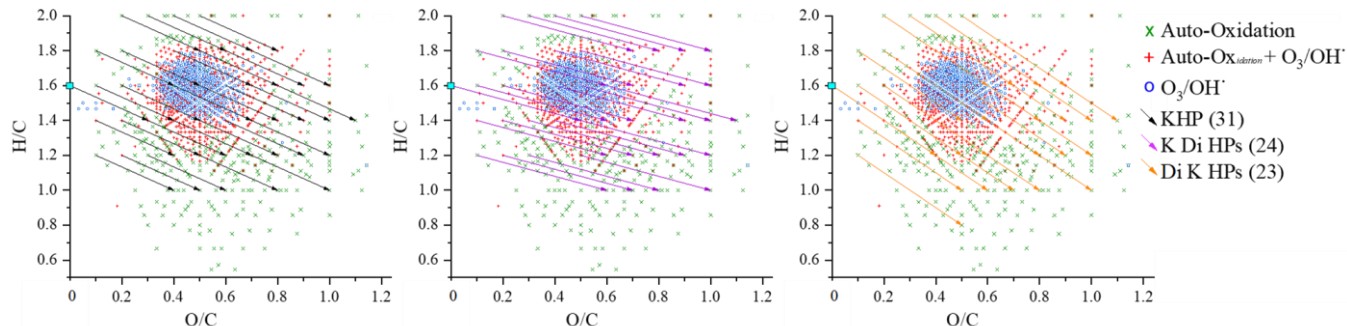

**Figure 9.** Representation in the van Krevelen diagram of the vectors associated with reaction mechanisms for the formation of KHPs, keto-di-hydroperoxides (K Di HPs), and di-ketohydroperoxides (Di K HPs).

## 5. Conclusion and perspectives

Numerous studies on the ozonolysis of limonene have allowed characterizing the reaction mechanisms of its oxidation by describing a large fraction of chemical products. In these mechanisms, the formation of a Criegee intermediate has often been described as the major pathway to oxidized compounds, associated with the more restricted formation of the OH˙ radical. Among these studies, some have shown that despite different oxidation conditions were used, implying differences in oxidation

mechanisms in ozonolysis and OH-oxidation/photolysis, many of the products were similar.

Our study suggest that in the absence of ozone, the oxidation by the OH˙ radical, common to ozonolysis, gives similar results in terms of chemical formulae of detected products. These results are in agreement with the previous study by Kourtchev et al (2015) which shows that the evolution of chemical species is mainly dominated by the concentration of OH radical. The present

study has allowed us to highlight autoxidation specific processes, such as formation of KHPs and diketones, occurrence of the
Korcek and Waddington reaction mechanisms.

The present results indicate that one should pay more attention to the Korcek and Waddington mechanisms yielding specific products observed here and in previous smog chamber experiments and field measurements.

Extensive oxidation of peroxy radicals yielding HOMs has been considered in atmospheric chemistry, but only recently a third-$O_2$ addition was added to combustion models, showing some influence on ignition modeling (Wang and Sarathy, 2016).
Here, limonene oxidation was initiated by reaction with molecular oxygen yielding alkyl radicals which form peroxy radicals by reaction with $O_2$. The oxidation proceeds further by sequential H-shift and $O_2$ addition yielding a wide range of products with odd numbers of O atoms ($C_{10}H_{14}O_{5,7,9,11}$). Besides, products with even numbers of O atoms were measured in this work ($C_{10}H_{14}O_{4,6,8,10}$). They are expected to come from the oxidation of limonene via the commonly accepted tropospheric oxidation mechanism forming alkoxy radicals, i.e., $RO_2^{\bullet} + RO_2^{\bullet} \rightarrow ROOOOR \rightarrow 2\,RO^{\bullet} + O_2$. The following sequential H-shift and $O_2$
addition on the alkoxy radicals yielded products of up to six $O_2$ addition in the present work ($C_{10}H_{14}O_{10}$). Such products have been reported in the previous studies considered here for comparison (Table 1). Although diverse experimental conditions were used here, in terms of concentration of reactants, temperature, reaction time, analysis conditions, we observed strong similitude in terms of molecular formulae detected in atmospheric and 'combustion' chemistry experiments. Besides, these two routes can produce a pool of OH radicals via decomposition of intermediates, e.g., $^{\bullet}OOQOOH \rightarrow {}^{\bullet}OH + HOOQ'O$ (KHP)
and $(HOO)_2Q'OO^{\bullet} \rightarrow {}^{\bullet}OH + (HOO)_2Q''O$ (keto dihydroperoxide) for the 'combustion' route and $^{\bullet}OOQOH \rightarrow {}^{\bullet}OH + OQ'OH$ (keto alcool) and $HOOQ'(OH)OO^{\bullet} \rightarrow {}^{\bullet}OH + HOOQ''(OH)O$ (keto hydroxy hydroperoxide) for the 'tropospheric' oxidation route. Furthermore, similarly to what has been reported in atmospheric chemistry studies (Witkowski and Gierczak, 2017;Jokinen et al., 2015;Walser et al., 2008;Kundu et al., 2012;Fang et al., 2017;Nørgaard et al., 2013;Bateman et al., 2009;Warscheid and Hoffmann, 2001;Hammes et al., 2019), a wide range of highly oxygenated products were detected, with
molecular formulae up to $C_{25}H_{32}O_7$ in the present work.

Analysis at the molecular level was complemented by observation at chemical family scale using Kendrick and van Krevelen visualization tools, necessary to compare and identify features in large data sets. Indeed, the formation of new HOMs and the development of combustion-related autoxidation are perfectly perceptible using these tools. The same is true for the oligomerization, which is not very important in autoxidation, in favor of addition and/or condensation reactions on limonene
that are prompt to increase the DBE. As it stands, the meshing within these chemical families according to reaction, thermodynamic, or kinetic criteria remains sketchy, but will certainly develop in the light of all the available experimental and theoretical inputs. The observed similarity in terms of chemical formulae obtained by different reaction mechanisms remains qualitative. It does not cover the aspects of quantification and chemical speciation, specific to a reaction mechanism. Nevertheless, we noticed that products similarity is predominant in ozonolysis and photolysis, whereas it is only close to 50%
for limonene low-temperature combustion. Indeed, lower similarity for the production of the lighter species, mostly observed in low-temperature combustion, was noticed. Further studies are needed to clarify the reasons for this difference and assess

the impact of residence time and/or ageing on the observed degree of similarity. Visualisation tools (e.g. VK diagrams, DBE plots) allowed to differentiate a number of the molecules that are likely related to the experimental conditions used in the current study (e.g. low temperature combustion). Among the chemical formulae observed in this work, some have not been reported in the studies considered here for comparison. It should be noted that other factors including experimental conditions (e.g. the use of flow tube reactor vs smog chambers) and/or MS instrument acquisition parameters (e.g. as demonstrated in the SI Figure 9) can be responsible for the observed differences with the compared studies. It would be interesting to perform additional experiments under conditions relevant to the atmosphere to verify that these chemical formulae are absent. Additional experiments in a JSR at lower initial temperatures and concentrations could also be undertaken to clarify the variation in product formation as temperature and reactant concentration change.

Further studies involving others terpenes are underway. They should confirm the results presented here.

## Acknowledgments

The authors gratefully acknowledge funding from Labex Caprysses (ANR-11-LABX-0006-01), the Labex Voltaire (ANR-10-LABX-100-01), and financial support from CPER and EFRD (PROMESTOCK and APPROPOR-e projects).

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
