# Peer review of "On the similarities and differences between the products of oxidation of hydrocarbons under simulated atmospheric conditions and cool-flames."

_Atmospheric Chemistry and Physics, 2020_

## Referee Comment (RC1) · Andrew Wozniak (Referee) · 14 Dec 2020

With "On the similarities and differences between the products of oxidation of hydrocarbons under simulated atmospheric conditions and cool-flames", Benoit et al. compare the molecular composition of limonene subjected to low temperature combustion conditions to literature data for limonene subjected to ozonolysis. The authors use the datasets to examine differences and similarities due to the reaction pathways and explore potential reaction mechanisms responsible for the observed composition. The dataset and interpretations are both very good quality, and the manuscript is techni-

cally sound. The justification for the study and its importance to understanding atmospheric chemistry are not well developed, however, and the manuscript requires revisions so readers can understand the importance of the work. These and other comments are discussed below. General Comments: At the end of the introduction (lines 67-75), the authors describe the study's aim as to compare the molecular composition of laboratory "cool flame" combustion of limonene to that of limonene subjected to atmospherically-relevant conditions. While this section describes the study aim/objective, what is needed is a justification/purpose for the study. How will comparing these two datasets provide valuable new information for our understanding of atmospheric organics (or human health, pollution, climate, etc.)? Just how these comparisons will benefit atmospheric chemists needs to be clearly stated so the study can be placed in proper context. Additionally, in the conclusions section (lines 391-407), the authors note that the composition of autoxidation processes are similar to those of ozonolysis and photooxidation, but they never make an argument for why the reader should care about these results. How this work enhances our understanding of atmospheric organic composition or processes in the atmosphere is never explained. The authors need to revise the Introduction and Conclusions sections to very clearly state the justification for and implications of the study. Additional Comments: - In section 2, "a.u." are used as units in multiple location. I am not familiar with these units. Please clarify. Line 89, please spell out FIA HESI/APCI for the reader who is unfamiliar with these acronyms. Line 112, Cite Kendrick (1963). Line 169, "Chemical formula with relative intensity was less than 1 ppm were not considered." To what does 1 ppm refer? Please explain how this relative intensity is calculated (relative to what? the highest magnitude peak? the total spectral magnitude?). In the Figure 2 caption, the circle symbols are not color coded as they are in the figure. Please fix or describe the color coding with words. Line 204, why were the auto oxidation experiments restricted to 2 s residence times? Can the short residence times relative to the ozonolysis and photo oxidation experiments explain the differences in composition? Figure 3 caption – the "aliphatics hydrocarbon," "aromatic hydrocarbon," and "unsaturated hydrocarbon"

compound classes classified exactly? Intuitively, the unsaturated hydrocarbon classification would refer to compounds having at least one double bond and would extend to higher H/C ratios. Aromatic hydrocarbons would be likely to show lower H/C ratios than merely unsaturated hydrocarbons. Please clarify how these classifications are calculated and use classification names that represent the probably compound structural characteristics. Figure 8 – the novelty and importance of these figures is overstated by the authors. Molecular formula exact mass datasets have been mathematically compared to identify reaction precursors and products in several previous studies (e.g., Gomez-Saez et al., 2016; Abdulla et al., 2020, and others). The same information can be visualized using Kendrick Mass defect analysis (using the expected difference(s) in elemental composition between precursor and product in place of CH2), vK diagrams, or other visualization techniques. The comparisons made in this instance are robust and valuable. The visualization and comparison are not as novel as stated by the authors.

---

## Referee Comment (RC2) · Anonymous Referee #2 · 18 Feb 2021

Review of "On the similarities and differences between the products of oxidation of hydrocarbons under simulated atmospheric conditions and cool flames" by Roland Benoit et al., (MS No.: acp-2020-1070).

Based on high resolution mass spectrometry (Orbitrap) analysis, the authors of this study compared the molecular characteristics of oxidation products of limonene in a jet-stirred reactor (JSR) to oxidation products of limonene by $^{\bullet}$OH and/or ozone under tropospheric relevant conditions in literature. The authors found that 771 of 1600 molecular formulae are common to both oxidation modes, while 462 molecular formulae are obtained solely by $^{\bullet}$OH and/or ozone oxidation, and 367 are produced in JCR experiments. On the basis of further analysis by ultra-high-pressure liquid chromatography (UHPLC), the authors investigated the potential occurrence of the Waddington mechanism and Korcek mechanism during the oxidation process of limonene in JSR. Even though the experiments are done carefully and the results are illustrated detailly, I am sceptical about the suitability and subsequent implications of the comparison of limonene oxidation under aforementioned conditions. It is significantly unclear about how relevant/representative of the applied/selected fuel lean condition to the real ambient conditions and whether the composition of the oxidation products in JCR largely dependent on the temperatures and limonene/$O_2$ concentration ratios. Finally, the manuscript is not written well, without presenting the novelty and atmospheric implications clearly. Therefore, I think the topic of this study fits better a combustion related journal (e.g., Combust. Flame.) rather than ACP.

---

## Referee Comment (RC3) · Anonymous Referee #3 · 22 Feb 2021

In the reviewed manuscript ("On the similarities and differences between the products of oxidation of hydrocarbons under simulated atmospheric conditions and cool flames", acp-2020-1070), the authors use ultrahigh-resolution mass spectrometry to compare the auto-oxidation products of limonene in a jet-stirred reactor with molecular formulae found in the literature for reactions with ozone and OH-radicals under more atmospherically relevant conditions. The study is technically sound, the manuscript is well written and it would be good to see the results published somewhere. However, the authors don't do a very good job arguing why this manuscript is appropriate for publication in ACP specifically. Maybe one could make the argument that if one has a very good understanding of the difference between the chemistry under the two very different conditions, one could use JSR experiments to predict atmospheric chemistry. . .but why would that be beneficial compared to just directly doing flow reactor or simulation chamber experiments? Given the strong focus on the compounds which aren't atmospherically relevant and the fact that only one set of conditions was tested for the JSR (so we have no idea about the breadth of variation in composition for JSR reactions), I wonder if this article wouldn't be better suited for a different journal (i.e. something combustion related).

Specific comments

Page 6, line 160: are these 1233 molecules or 1233 molecular formulae? For a fair comparison with your acquired data, it should be the latter. In addition, it is not completely clear from the text whether the list of 1233 compounds/molecular formulae only contains formulae that were common to all of the "atmospheric" studies or every formula found in any of the studies. If it is the latter, I think some caveats regarding chemical diversity should be added since precursor concentration can affect the product composition and many of the listed studies are quite far away from atmospheric concentrations in this regard.

Page 7. Line 169: relative to what? I'm assuming it is supposed to be relative to the peak maximum of the highest mass peak in the spectrum, but this should be stated explicitly.

Page 10, line 249: could you elaborate a bit more about how you arrived at your compound family classification here? Especially since the limits you are setting seem to differ from the cited Bianco et a. regarding e.g. the aromatic structures.
* * *

---

## Author Comment (AC2) · 25 Feb 2021

**acp-2020-1070-RC2-ANSWERS**

**Comment** : Even though the experiments are done carefully and the results are illustrated detailly, I am sceptical about the suitability and subsequent implications of the comparison of limonene oxidation under aforementioned conditions.

**Answer**: We consider this comparison of great importance because it demonstrates strong and unexpected similitude in terms of reaction pathways to products of autoxidation and OH/O3 initiated oxidation. To our knowledge, this is the first time these results are reported, not only for limonene, but for any other organic compound.

**Comment**: It is significantly unclear about how relevant/representative of the applied/selected fuel lean condition to the real ambient conditions and whether the composition of the oxidation products in JCR largely dependent on the temperatures and limonene/$O_2$ concentration ratios.

**Answer**: The reviewer missed pour point: demonstrating the strong and unexpected similitude in terms of reaction pathways to products of autoxidation and OH/O3 initiated oxidation of limonene.

**Comment**: Finally, the manuscript is not written well, without presenting the novelty and atmospheric implications clearly. Therefore, I think the topic of this study fits better a combustion related journal (e.g., Combust. Flame.) rather than ACP.

**Answer**: The revision attempts to address this critic by emphasizing the novelty of this work: "To our knowledge, this is the first time this important finding is reported. It goes beyond expectations." Was added to the Conlusion section. However, this paper does not pretend to revolutionize the understanding of atmospheric processes but rather point out that both atmospheric chemists and combustion chemist can learn from their studies performed under different initial conditions. Regarding the choice of the journal to publish our work, we believe ACP is a better choice. One of the authors has been Editor in Chief of Combustion and Flame. He knows the present work would not be publishable in Combustion and Flame. In fact, in combustion the processes yielding highly oxygenated products are neglected and not included in the most recent chemical kinetic reaction mechanisms. Only very recently the so-called 'third O2 addition' has been considered.

---

## Author Comment (AC1)

| Reviewer's comments | Answers and modifications of the manuscript |
|---|---|
| With "On the similarities and differences between the products of oxidation of hydrocarbons under simulated atmospheric conditions and cool-flames", Benoit et al. Compare the molecular composition of limonene subjected to low temperature combustion conditions to literature data for limonene subjected to ozonolysis. The authors use the datasets to examine differences and similarities due to the reaction pathways and explore potential reaction mechanisms responsible for the observed composition. The dataset and interpretations are both very good quality, and the manuscript is techni- cally sound. The justification for the study and its importance to understanding atmospheric chemistry are not well developed, however, and the manuscript requires revisions so readers can understand the importance of the work. These and other comments are discussed below. | |
| General Comments:

At the end of the introduction (lines 67-75), the authors describe the study's aim as to compare the molecular composition of laboratory "cool flame" combustion of limonene to that of limonene subjected to atmospherically-relevant conditions. While this section describes the study aim/objective, what is needed is a justification/purpose for the study. How will comparing | The introduction was modified to address the reviewer's comments :

« Introduction »

Several corrections
 L 26-29 : Air particulates are responsible for increasing death rates and deseases (Lim et al., 2012) and have a negative impact on climate (Myhre et al., 2013). With increasing temperatures, particularly in summer, biogenic emissions of volatile organic compounds are more important than anthropogenic (Lamarque et al., 2010). Among them, one finds terpenes emitted by vegetation. They represent …

L-46 : Whereas in atmospheric chemistry peroxy radicals self- and cross-reactions are very important (Wallington et al., 1992), in … |

| | |
|---|---|
| hese two datasets provide valuable new information for our understanding of atmospheric organics (or human health, pollution, climate, etc.)? Just how these comparisons will benefit atmospheric chemists needs to be clearly stated so the study can be placed in proper context. | L 73-79 : Can atmospheric chemistry benefit from combustion chemistry studies and vice versa? Litterature seems to indicate that cool flame combustion presents oxidation mechanisms similar to those observed in biomass burning (Koppmann et al., 2005;Reid et al., 2004). Koppmann et al. study showed that the rise in the temperature of the biomass causes the evaporation of all the families of VOCs (0.2 to 15% by mass of the biomass) as well as their "combustion without flame". This "combustion" consists in an oxidation of VOCs more or less pronounced, depending on the temperature and humidity of the environment, and contributes to the formation of a wide diversity of chemical compounds.

L 86-88 : Inventory and chemical speciation of oxidation products, as well as the comparison with products of other modes of oxidation (ozonolysis, OH° and photolysis) should yield a better characterization of the specificities of each mode of oxidation and provide new targetsdata for field experiments. |
| Additionally, in the conclusions section (lines 391-407), the authors note that the composition of autoxidation processes are similar to those of ozonolysis and photooxidation, but they never make an argument for why the reader should care about these results. How this work enhances our understanding of atmospheric organic composition or processes in the atmosphere is never explained. | The conclusion was modified to address the reviewer's comments :

L 413-419 : We notifieed that in the absence of ozone, the oxidation by the OH˙ radical, common to ozonolysis, gives similar results. Nevertheless, this study has allowed us to highlight auto-oxidation specific processes, such as formation of KHPs and diketones, occurrence of the Korcek and Waddington reaction mechanisms. Extensive oxidation of peroxy radicals yielding HOMs is considered in atmospheric chemistry only recently; it was added to combustion models, showing some influence on ignition modeling (Wang and Sarathy, 2016). The present results indicate that one should pay more attention to the Korcek and Waddington mechanisms yielding specific products observed here and in previous smog chamber experiments and field measurements. |
| The authors need to revise the Introduction and Conclusions sections to very clearly state the justification for and implications of the study. | L426-432 : The observed similarity in terms of chemical compounds obtained by different reaction mechanisms remains qualitative. It does not cover the aspects of quantification and chemical speciation, specific to a reaction mechanism. Nevertheless, we noticed that products similarity is predominant in ozonolysis and photolysis, whereas it is only close to 50% for limonene low-temperature combustion. Lower similarity for the production of the lighter species, mostly observed in low-temperature combustion, was noticed. Further studies are needed to |

clarify the reasons for this difference and assess the impact of residence time and/or ageing on the observed degree of similarity. In any case, it appears possible to differentiate, thanks to visualization tools, a signature of the compounds formed by low-temperature combustion.

Additional Comments:

- In section 2, "a.u." are used as units in multiple location. I am not familiar with these units. Please clarify.

"a.u." is spelled out in the revision: "arbitrary units (a.u.)" Line 89

Line 89, please spell out FIA HESI/APCI for the reader who is unfamiliar with these acronyms.

Line 89, we spell out FIA HESI/APCI in the revision: "flow injection analyses heated electrospray ionization/atmospheric pressure chemical ionization (FIA HESI/APCI)"

Line 112, Cite Kendrick (1963).

Reference cited." A Mass Scale Based on CH2 = 14.0000 for High Resolution Mass Spectrometry of Organic Compounds. Edward. Kendrick Anal. Chem. 1963, 35, 13, 2146–2154. https://doi.org/10.1021/ac60206a048

Line 169, "Chemical formula with relative intensity was less than 1 ppm were not considered." To what does 1 ppm refer? Please explain how this relative intensity is calculated (relative to what? the highest magnitude peak? the total spectral magnitude?).

One should read "Chemical formula with relative intensity less than 1 ppm (relative to the highest magnitude peak) were not considered."

In the Figure 2 caption, the circle symbols are not color coded as they are in the figure. Please fix or describe the color coding with words.

Corrected

| | |
|---|---|
| Line 204, why were the auto oxidation experiments restricted to 2 s residence times? Can the short residence times relative to the ozonolysis and photo oxidation experiments explain the differences in composition? | To operate our JSR under ideal 'perfectly stirred reactor' conditions, one must respect a range of residence time compatible with reactor geometry. Our reactor cannot operate at longer residence time at atmospheric pressure. This is a technical limitation. Observed differences could be ascribed to differences in residence times. |
| Figure 3 caption – the "aliphatics hydrocarbon," "aromatic hydrocarbon," and "unsaturated hydrocarbon" compound classes classified exactly? Intuitively, the unsaturated hydrocarbon classification would refer to compounds having at least one double bond and would extend to higher H/C ratios. Aromatic hydrocarbons would be likely to show lower H/C ratios than merely unsaturated hydrocarbons. Please clarify how these classifications are calculated and use classification names that represent the probably compound structural characteristics. | The border between unsaturated and aromatic compounds is narrow. These compounds have a lower H/C ratio than aliphatic compounds. The spaces cited in this work are from the literature (Bianco et al., 2018; Nozière et al., 2015). There is obviously an overlap of these two spaces which nevertheless can be positioned with respect to the chemistry studied. For an increasing number of carbon (with C>6), the H/C ratio will be preferentially higher for aromatics compared to unsaturated chains.
[Figure]
 |
| Figure 8 – the novelty and importance of these figures is overstated by the authors. Molecular formula exact mass datasets have been mathematically compared to identify reaction precursors and products in several previous studies (e.g., Gomez-Saez et al., 2016; Abdulla et al., 2020, and others). The same information can be visualized using Kendrick Mass defect analysis (using the expected difference(s) in elemental composition between | The reviewer said that "The visualization and comparison are not as novel as stated by the authors." The notion of screening integrates here an algorithm allowing in this van Krevelen space to quantify the most probable chemical reactions. The exploratory field is more advanced and easier to interpret than many representations of Kendricks. It is also possible to specify in the algorithm restrictive conditions for the evolution of the DBE allowing a more accurate analysis than the one described in the literature. |

| precursor and product in place of CH2), vK diagrams, or other visualization techniques. The comparisons made in this instance are robust and valuable. The visualization and comparison are not as novel as stated by the authors. | |

| OTHER CORRECTIONS : |
| --- |
| L.111: 'CAN' replaced by 'ACN' |
| L.127: reference added : Kendrick 1963 |
| L.265: 'aliphatics' replaced by 'aliphatic' |
| L 279: New chemical reaction |
| L. 314 reference added : Mutzel et al. 2015 "Jensen et al. 1981" |
| L. 365 : corrected chemical formula indices |
| Figure 8 '+ Auto-Ox$_{idation}$' corrected |

---

## Author Response (AR1)

We thank all the reviewers for their constructive comments and suggestions which allowed us to improve our article.

We have provided detailed responses (in blue) to each of the comments below. The modifications of the original text are identified in red.

**acp-2020-1070-RC1-ANSWERS**

**General Comments:**

**At the end of the introduction (lines 67-75), the authors describe the study's aim as to compare the molecular composition of laboratory "cool flame" combustion of limonene to that of limonene subjected to atmospherically-relevant conditions. While this section describes the study aim/objective, what is needed is a justification/purpose for the study. How will comparing these two datasets provide valuable new information for our understanding of atmospheric organics (or human health, pollution, climate, etc.)? Just how these comparisons will benefit atmospheric chemists needs to be clearly stated so the study can be placed in proper context.**

**Additionally, in the conclusions section (lines 391-407), the authors note that the composition of autoxidation processes are similar to those of ozonolysis and photooxidation, but they never make an argument for why the reader should care about these results. How this work enhances our understanding of atmospheric organic composition or processes in the atmosphere is never explained.**

**The authors need to revise the Introduction and Conclusions sections to very clearly state the justification for and implications of the study**

**Answer**:

The Abstract, Introduction and the Conclusion were revised and the link of this work to atmospheric chemistry is better presented in the revision. A focus on the comparison of these two chemistries, highlighting the common points and the differences, was carried out in the result section.

We would like to point out that under our experimental conditions, which are different from those found in the troposphere, we observe strong similitude in terms of detected molecular formulae of products. We clarified in the revision that both (i) autoxidation (generally considered as a combustion process) and (ii) reaction pathways (successive H-shift and molecular oxygen addition) following the initial formation of alkoxy radicals via the accepted mechanism of oxidation of VOCs in the troposphere, i.e., ROO + ROO → ROOOOR → 2 RO + $O_2$ (ROO + NO → RO + $NO_2$ is not relevant in our conditions where no nitric oxide is present) occur in our JSR, based on detected molecular formulae. Our findings are rather unexpected. The fact that these oxidation routes occur under very different conditions need to be brought to the attention of physical chemist involved in atmospheric and combustion related works.

**The corrected section reads as follows:**

[revised manuscript text omitted]

**Additional Comments:**

**- In section 2, "a.u." are used as units in multiple location. I am not familiar with these units. Please clarify.**
**Line 89, please spell out FIA HESI/APCI for the reader who is unfamiliar with these acronyms.**

**Answer**: "a.u." is spelled out in the revision: "arbitrary units (a.u.)" Line 89

Line 89, we spell out FIA HESI/APCI in the revision: "flow injection analyses heated electrospray ionization/atmospheric pressure chemical ionization (FIA HESI/APCI)"

**Line 112, Cite Kendrick (1963).**
**Answer**: Reference cited." A Mass Scale Based on CH2 = 14.0000 for High Resolution Mass Spectrometry of Organic Compounds. Edward. Kendrick, Anal. Chem. 1963, 35, 13, 2146–2154. https://doi.org/10.1021/ac60206a048

**Line 169, "Chemical formula with relative intensity was less than 1 ppm were not considered." To what does 1 ppm refer? Please explain how this relative intensity is calculated (relative to what? the highest magnitude peak? the total spectral magnitude?).**

**Answer**: One should read "Chemical formula with relative intensity less than 1 ppm (relative to the highest magnitude peak) were not considered."

**In the Figure 2 caption, the circle symbols are not color coded as they are in the figure. Please fix or describe the color coding with words.**

Corrected

**Line 204, why were the auto oxidation experiments restricted to 2 s residence times? Can the short residence times relative to the ozonolysis and photo oxidation experiments explain the differences in composition?**

**Answer**: To operate our JSR under ideal 'perfectly stirred reactor' conditions, one must respect a range of residence time compatible with reactor geometry. Our reactor cannot operate at longer residence time at atmospheric pressure. This is a technical limitation. Observed differences could be ascribed to differences in residence times.

**Figure 3 caption – the "aliphatics hydrocarbon," "aromatic hydrocarbon," and "unsaturated hydrocarbon" compound classes classified exactly? Intuitively, the unsaturated hydrocarbon classification would refer to compounds having at least one double bond and would extend to higher H/C ratios. Aromatic hydrocarbons would be likely to show lower H/C ratios than merely unsaturated hydrocarbons. Please clarify how these classifications are calculated and use classification names that represent the probably compound structural characteristics.**

**Answer**: There is an inversion on the legends b and c of figure 3. The correct legend is (b) unsaturated hydrocarbons; (c) aromatics hydrocarbons; this is corrected in the revision.

**Figure 8 – the novelty and importance of these figures is overstated by the authors. Molecular formula exact mass datasets have been mathematically compared to identify reaction precursors and products in several previous studies (e.g., Gomez-Saez et al., 2016; Abdulla et al., 2020, and others). The same information can be visualized using Kendrick Mass defect analysis (using the expected difference(s) in elemental composition between precursor and product in place of CH2), vK diagrams, or other visualization techniques. The comparisons made in this instance are robust and valuable. The visualization and comparison are not as novel as stated by the authors.**

Answer: The notion of novelty has been removed. It is only a question of improvements.

Thanks to all the tools used in this work, especially the last one, the screening of selected classes of compounds is considerably simplified, and the mesh constituted by the reaction vectors represents an efficient method for evaluating the nature of a mixture of oxidation products.
* * *
**acp-2020-1070-RC2-ANSWERS**

**Comment : Even though the experiments are done carefully and the results are illustrated detailly, I am sceptical about the suitability and subsequent implications of the comparison of limonene oxidation under aforementioned conditions.**

**It is significantly unclear about how relevant/representative of the applied/selected fuel lean condition to the real ambient conditions and whether the composition of the oxidation products in JCR largely dependent on the temperatures and limonene/O2 concentration ratios.**

**Answer**:
The oxidation conditions were adapted to the observation of ketohydroperoxides and HOMs. These chemical species present in atmospheric chemistry as well as in combustion remain difficult to observe and even more difficult to characterize. In order to overcome this limitation and to identify these species by fragmentation, we adjusted the initial concentration of limonene and used the short residence time specific to JSR (which is continuous-flow tank reactor). A JSR is a homogeneous system, very useful for studying chemical kinetics of oxidation processes. Regarding the products we focused on, such as ketohydroperoxides, one must keep in mind these are key-intermediates in both combustion (cool-flames) and tropospheric chemistry. The review of Bianchi (Chem. Rev. 119 (2019) 3472; see section 4.1) emphasizes the importance of that chemistry in the atmosphere. Due to the relative simplicity of ketohydroperoxides compare to HOMs, we devoted more attention on their characterization. They represent the early steps of oxidation of fuels or VOCs. We went beyond the measurement of ketohydroperoxides. The products of alkoxy radicals oxidation (ROO + ROO $\rightarrow$ ROOOOR $\rightarrow$ 2 RO + O$_2$ followed by multiple H-shits and molecular oxygen addition forming highly oxygenated products) were also measured showing the occurrence of this reaction pathway under our experimental conditions, although somewhat unexpected.
The abstract, the introduction and conclusion were modified to address the reviewer's comments.
The framework of the study has been justified with respect to atmospheric chemistry and the objective clarified.

**The corrected section reads as follows:**

[revised manuscript text omitted]

*Limonene*       (R$^\bullet$)     (RO$_2$$^\bullet$)                                 (RO$^\bullet$)         (Q$^\bullet$OH)

$$C_{10}H_{14}OH \xrightarrow[\text{(2)}]{+O_2} C_{10}H_{15}O_3 \xrightarrow[\text{+O}_2 \text{ (3)}]{\text{H-shift}} C_{10}H_{15}O_5 \xrightarrow[\text{+O}_2 \text{ (4)}]{\text{H-shift}} C_{10}H_{15}O_7 \xrightarrow[\text{+O}_2 \text{ (5)}]{\text{H-shift}} C_{10}H_{15}O_9 \xrightarrow[\text{+O}_2 \text{ (6)}]{\text{H-shift}} C_{10}H_{15}O_{11}$$

(Q$^\bullet$OH)     $^\bullet$OOQOH     HOOQ'(OH)OO$^\bullet$    (HOO)$_2$Q''(OH)OO$^\bullet$   (HOO)$_3$P(OH)OO$^\bullet$   (HOO)$_4$P'(OH)OO$^\bullet$

Each of the above with downward arrows labeled $-$OH and H-shift leading to:

$$C_{10}H_{14}O_2 \quad C_{10}H_{14}O_4 \quad C_{10}H_{14}O_6 \quad C_{10}H_{14}O_8 \quad C_{10}H_{14}O_{10}$$

OQ'OH     HOOQ''(OH)O     (HOO)$_2$P(OH)O     (HOO)$_4$P'(OH)O     (HOO)$_6$P''(OH)O

(b)

$$R^\bullet \xrightarrow[\text{(1)}]{+O_2} ROO^\bullet \xrightarrow{\text{H-shift}} QOOH \xrightarrow[\text{(2)}]{+O_2} {}^\bullet OOQOOH \xrightarrow{\text{H-shift}} HOOQ'OOH \xrightarrow[\text{(3)}]{+O_2} (HOO)_2Q'OO^\bullet$$

QOOH → (down, $-$OH / H-shift) → HOOQ'O

(HOO)$_2$Q'OO$^\bullet$ → (down, $-$OH / H-shift) → (HOO)$_2$Q''O

$$(HOO)_2Q'OO^\bullet \xrightarrow{\text{H-shift}} (HOO)_2Q''OOH \xrightarrow[\text{(4)}]{+O_2} (HOO)_3Q''OO^\bullet \xrightarrow{\text{H-shift}} (HOO)_3POOH \xrightarrow[\text{(5)}]{+O_2} (HOO)_4POO^\bullet$$

(HOO)$_3$Q''OO$^\bullet$ → (down, $-$OH / H-shift) → (HOO)$_3$PO

(HOO)$_4$POO$^\bullet$ → (down, $-$OH / H-shift) → (HOO)$_4$P'O

[revised manuscript text omitted]
", acp-2020-1070), the authors use ultrahigh-resolution mass spectrometry to compare the auto-oxidation products of limonene in a jet-stirred reactor with molecular formulae found in the literature for reactions with ozone and OH-radicals under more atmospherically relevant conditions. The study is technically sound, the manuscript is well written and it would be good to see the results published somewhere. However, the authors don't do a very good job arguing why this manuscript is appropriate for publiC1 ACPD Interactive comment Printer-friendly version Discussion paper cation in ACP specifically. Maybe one could make the argument that if one has a very good understanding of the difference between the chemistry under the two very different conditions, one could use JSR experiments to predict atmospheric chemistry. . .but why would that be beneficial compared to just directly doing flow reactor or simulation chamber experiments? related).**

**Answer**:
The Introduction and the Conclusion were revised and the link of this work to atmospheric chemistry is better presented in the revision. A focus on the comparison of these two chemistries, highlighting the common points and the differences, was carried out in the result section.
We do not claim using a JSR (which is continuous-flow tank reactor) can replace smog chambers or flow-tube reactors. All of these systems can bring useful information. What we claim is that under our experimental conditions, which are different from those found in the troposphere, we observe strong similitude in terms of detected molecular formulae of products. We clarified in the revision that both (i) autoxidation (generally considered as a combustion process) and (ii) reaction pathways (successive H-shift and molecular oxygen addition) following the initial formation of alkoxy radicals via the accepted mechanism of oxidation of VOCs in the troposphere, i.e., ROO + ROO → ROOOOR → 2 RO + $O_2$ (ROO + NO → RO + $NO_2$ is not relevant in our conditions where no nitric oxide is present) occur in our JSR, based on detected molecular formulae. Our findings are rather unexpected. The fact that these oxidation routes occur under very different conditions need to be brought to the attention of physical chemist involved in atmospheric and combustion related works.

**Given the strong focus on the compounds which aren't atmospherically relevant and the fact that only one set of conditions was tested for the JSR (so we have no idea about the breadth of variation in composition for JSR reactions), I wonder if this article wouldn't be better suited for a different journal (i.e. something combustion)** ".

**Answer**:
We agree that all the parameters which can be varied in a JSR were not. One could argue this is the same in many studies performed at room temperature in flow tube reactor or a smog chamber. At least, a JSR is a

homogeneous system, very useful for studying chemical kinetics of oxidation processes. Regarding the products we focused on, such as ketohydroperoxides, one must keep in mind these are key-intermediates in both combustion (cool-flames) and tropospheric chemistry. The review of Bianchi (Chem. Rev. 119 (2019) 3472; see section 4.1) emphasizes the importance of that chemistry in the atmosphere. Due to the relative simplicity of ketohydroperoxides compare to HOMs, we devoted more attention on their characterization. They represent the early steps of oxidation of fuels or VOCs. We went beyond the measurement of ketohydroperoxides. The products of alkoxy radicals oxidation (ROO + ROO → ROOOOR → 2 RO + $O_2$ followed by multiple H-shits and molecular oxygen addition forming highly oxygenated products) were also measured showing the occurrence of this reaction pathway under our experimental conditions, although somewhat unexpected.

Regarding the choice of the journal, ACP (or another atmosphere-oriented journal) looks to be a good place for publishing our results. A combustion journal would most likely not be interested because the processes yielding highly oxygenated products would be viewed as marginal and worth publishing in an atmosphere-oriented journal.

**The corrected section reads as follows:**

[revised manuscript text omitted]
 ⇆ R + $H_2O$, R+ $O_2$ ⇆ ROO, ROO ⇆ QOOH, QOOH + $O_2$ ⇆ OOQOOH, OOQOOH ⇆ HOOQ'OOH ⇆ HOOQ'O + OH, HOOQ'O ⇆ OQ'O + OH. However, recent studies reported the formation of HOMs during the so-called low-temperature oxidation (500–600 K) of hydrocarbons and other organics, e.g., alcohols, aldehydes, ethers, esters (Wang et al., 2018;Wang et al., 2017b;Belhadj et al., 2020). There, the H-atom transfer in the

OOQOOH intermediate does not involve the H-C-OOH group but another H-C group, opening new oxidation pathways. Such alternative pathways do not yield ketohydroperoxides, and a third $O_2$ addition to HOOQ'OOH yielding OOQ'(OOH)$_2$ can occur. This sequence of reactions can proceed again, yielding highly oxygenated products (Wang et al., 2017b;Belhadj et al., 2020;Belhadj et al., 2021). Also, QOOH can decompose via: QOOH → OH + cyclic ether, QOOH → OH + carbonyl + olefin, and QOOH → $HO_2$ + olefin. In few studies devoted to the understanding of atmospheric oxidation mechanism of hydrocarbons yielding highly oxidized products, autoxidation was proposed as a pathway to organic aerosols, e.g. (Jokinen et al., 2014a;Jokinen et al., 2015;Mutzel et al., 2015;Berndt et al., 2016;Crounse et al., 2013;Ehn et al., 2014). The early H-shift, ROO ⇆ QOOH, is favored by increased temperature, which explains its importance in autoignition, but the presence of substituents such as OH, C=O, and C=C in the ROO radical can significantly increase the rate of H-shift making it of significance at atmospheric temperatures (Bianchi et al., 2019).

Beside these processes, the Waddington mechanism (Ray et al., 1973), involving OH and $O_2$ successive additions on a C=C double bond, followed by H-atom transfer from –OH to –OO, can occur, yielding carbonyl compounds: R-C=C-R' + OH ⇆ R-C-C(-R')-OH, R-C-C(-R')-OH + $O_2$ ⇆ OO-C(-R)-C(-R')-OH ⇆ HOO-C(-R)-C(R')-O ⇆ OH + R-C=O + R'-C=O. The Korcek mechanism (Jensen et al., 1981) through which γ-ketohydroperoxides are transformed into a carboxylic acid and a carbonyl compound can occur too. The formation of carboxylic acids and carbonyl products via the Korcek mechanism has already been postulated by Mutzel et al. (Mutzel et al., 2015) whereas it is frequently considered in recent kinetic combustion modeling (Ranzi et al., 2015).

Then, questions arise: what are the similarities and differences between the products of oxidation of hydrocarbons under simulated atmospheric conditions and cool-flames? Do oxidation routes observed in autoxidation (cool flames) play a significant role under atmospheric conditions? Can atmospheric chemistry benefit from combustion chemistry studies and vice versa?

~~Litterature seems to indicate that cool flame combustion presents oxidation mechanisms similar to those observed in biomass burning . Koppmann et al. study showed that the rise in the temperature of the biomass causes the evaporation of all the families of VOCs (0.2 to 15% by mass of the biomass) as well as their "combustion without flame". This "combustion" consists in an oxidation of VOCs more or less pronounced, depending on the temperature and humidity of the environment, and contributes to the formation of a wide diversity of chemical compounds.~~

[revised manuscript text omitted]

**Comment: Page 6, line 160: are these 1233 molecules or 1233 molecular formulae? For a fair comparison with your acquired data, it should be the latter.**

**Answer**: One should read '1233 molecular formulae'; this is corrected in the revision

**Comment: In addition, it is not completely clear from the text whether the list of 1233 compounds/molecular formulae only contains formulae that were common to all of the "atmospheric" studies or every formula found in any of the studies. If it is the latter, I think some caveats regarding chemical diversity should be added since precursor concentration can affect the product composition and many of the listed studies are quite far away from atmospheric concentrations in this regard.**

**Answer**: The list of 1233 compounds/molecular formulae corresponds to a set of every formula found in any of the studies found in the literature. This is corrected in the revision.
To address the second point, we revised the statement in question: "These nine experimental studies performed under diverse initial conditions, as shown in Table 1, yielded a first set of 1233 molecular formulae for an inventory which, although incomplete, gives a broad representativeness of the chemical products which can result from limonene ozonolysis and OH-initiated photooxidation.

**Comment: Page 7. Line 169: relative to what? I'm assuming it is supposed to be relative to the peak maximum of the highest mass peak in the spectrum, but this should be stated explicitly.**

**Answer**: It was stated "*Chemical formula with relative intensity was less than 1 ppm were not considered*". To address this point, the revised sentence reads: "Chemical formula with relative intensity to the peak maximum of the highest mass peak in the spectrum less than 1 ppm were not considered."

**Comment: Page 10, line 249: could you elaborate a bit more about how you arrived at your compound family classification here? Especially since the limits you are setting seem to differ from the cited Bianco et a. regarding e.g. the aromatic structures.**

**Answer**: There is an inversion on the legends b and c of figure 3. The correct legend is (b) unsaturated hydrocarbons; (c) aromatics hydrocarbons; this is corrected in the revision.

---

## Editor Decision (ED1)

Dear Dr Benoit,

Thank you for posting additional responses to the reviewers (on the 26th of March). I also would like to thank you for performing additional experiments with varying HRMS instrument parameters. I suggest these results are to be added into Supplementary Information (SI).

With regards to your response that "*you observed that the change in acquisition mass range (50-750 to 50-450 m/z) had an effect only on the quantitative aspect of the results, without changing the set of chemical molecules identified.*"

This is well expected, as the ion response depends on the RF amplitude applied to the c-trap (in Orbitrap). This effect strongly depends on the first mass of the mass range (e.g. *m/z* 50 in your case). The lower the starting mass, the lower the RF amplitude. With a low RF amplitude, higher masses are not trapped/transmitted so efficiently. As a rule of thumb, the c-trap can catch a mass range of "starting mass - (starting mass*15)". I expect this difference to be substantial when comparing *m/z* 50-750 vs *m/z* 150-750. So, I fail to understand the reason for comparing data from scan ranges of *m/z* 50-750 to *m/z* 50-450 and I am not surprised at all that your observed differences at those conditions were minor.

I either suggest performing additional experiments with applying the following scan ranges *m/z* 50-750 vs *m/z* 150-750 and adding your results to SI as a proof of concept or adding a statement in the text with a caveat for comparison of results from various publications. So that the reader, who is not familiar with HRMS technique, can understand that the molecular formulae that are only present in your study (or vice versa) could also be due to the differences in the MS acquisition parameters.

With regards to your following response:

*"The HESI source was used only to compare the chemical formulas of different works. It is preferably adapted to a wide range of mass detection. Given the set of experimental differences you recalled, one might have expected several distinct sets of chemical formulas that are usually well visualized with Kendrick or van Krevelen type graphical tools. But, it is a continuity and an important similarity of data that we observed. The comparison is only qualitative and the range of masses studied 50-750 is relatively small. The differences observed in HESI are often due to a bad optimization of the experimental conditions. A bad pot ential difference, a too high injection quantity, a temperature not adapted to the studied elements, an initially not properly cleaned ionization chamber, a too high nitrogen flow,... Unfortunately, even with bad settings, the ionization can occur giving false results. We gave a lot of attention to all these parameters and compared our results to those obtained in APCI. Given our sensitivity, we did not find any qualitative difference. Concerning the existence of compounds through accretion reactions, it is difficult to exclude it definitively, but our observations did not allow us to highlight its presence."*

It is not a matter of 'bad' or 'good' settings (as you mentioned in your responses), it is to what compounds your system was optimised to. I realise you are comparing molecular formulae, but those formulae are determined from ions which transmission is affected by the MS instrument settings. Again, I don't see any problem if you use the same system and the same HRMS settings to compare your numerous experiments; however, when comparing them with external data, such caveat needs to be stated in the paper as ACP is not a Mass Spectrometry specialised journal and such caveat is not apparent for those who do not use this technique.

I agree with your statement: "*We cannot exclude that some new compounds could be associated with unrealistic compounds compared to the atmosphere,….*" However, I also think this needs to be emphasised in your manuscript, especially considering the two reviewers' comments on the relevance of the presented work to the real atmospheric conditions.

---

## Author Response (AR2)

Dear Dr Kourtchev,

Thank you for your comments and suggestions. We have taken them into account in the revised manuscript (March 31). We have added two corrections and a reference (Hecht et al., 2019).

The first one in the text line 169, page 6 and the second one in the supplementary material, Figure S9 (additional experimental results with different acquisition mass ranges as you suggested).
With these updates, we hope the manuscript will be suitable for publication in your Journal

Best regards,

Roland BENOIT

---

## Author Response (AR3)

Dear Dr Kourtchev,

Thank you for your comments and suggestions. We have taken them into account in the revised manuscript (April 2).

In the second part of this document, you will find the previous changes since October 14, 2020.

Best regards,

**Roland BENOIT**

**Comments :** I think you unintentionally combined responses on two comments into one. The way it can be read now that RF voltages will impact on observation of atmospherically unrealistic compounds. I assume this is not what you wanted to say. It is highly unlikely that RF voltages can lead to atmospherically unrealistic compounds. Although some MS conditions have shown to lead to formation of atmospherically irrelevant compounds (e.g. formation of non-covalent artifacts), these would not be related to RF voltages (please see my second comment below).

**Here are the changes we have made**

**Line 171-180, page 6 (Experiments)**

Nevertheless, it should be considered that some of the molecules presented in this study could result from our experimental conditions (continuous-flow tank reactor, concentration of reagents, temperature, reaction time,...) and to some extent to our acquisition conditions, different from those in the cited studies (Table 1). Indeed, the use of a continuous-flow tank reactor operating at elevated temperature, as well as a high initial concentration of reactants can induce the formation of unrealistic atmospheric compounds. About the acquisition conditions, the selected mass range has an influence on the transmission factor, especially at the higher mass range. Figure 9s, in the supplementary material, compares two spectra of oxidized limonene with different acquisition mass ranges. A decrease in trapping efficiency at higher masses is clearly visible when changing the mass range from m/z 150-750 to 50-750. It is also necessary to consider the possible formation of non-covalent artifacts, without excluding an incidence on the DBE number. A more detailed description of these technical aspects is available in a recent review (Hecht et al., 2019).

**Dr Kourchev**

**Comments**: Atmospherically unrealistic compounds would mainly arise from the experimental conditions, e.g. use of high concentrations of hydrocarbons in the flow tube vs more atmospherically relevant concentrations used in the chamber experiments (as stated by two reviewers). This needs to be clearly stated in your manuscript and in the right place (e.g. in introductory and conclusion sessions) to address the comment of the two reviewers. I would also add a statement of caution when you compare/consider the molecular composition or chemical evolution from different experiments in your Table 1 (chamber vs flow tube, and different MS instruments).

**Here are the changes we have made**

Line 24—25, page 1 (Abstract)

Unexpectedly, because of diversity of experimental conditions in terms of continuous-flow tank reactor, concentration of reactants, temperature, reaction time, mass spectrometry techniques and analyses conditions, the results indicate that among the 1138 presently detected products, many oxygenates found in earlier studies of limonene oxidation by OH and/or ozone are also produced under the present conditions.

**Line 531-534, page 24 (Conclusion and perspectives)**

Although diverse experimental conditions were used here, in terms of concentration of reactants, temperature, reaction time, analysis conditions, we observed strong similitude in terms of molecular formulae detected in atmospheric and 'combustion' chemistry experiments.

**Line 555-559, page 25 (Conclusion and perspectives)**

Among the chemical formulas observed in this work, some had not been reported in the 9 atmosphereoriented studies considered here for comparison. It would be interesting to perform additional experiments under conditions relevant to the atmosphere to verify that these chemical formulas are absent. Additional experiments in a JSR at lower initial temperatures and concentrations could also be undertaken to clarify the variation in product formation as temperature and reactant concentration change.

**Dr Kourchev**

**Comments** : The use of MS acquisition scan ranges, on the other hand, can affect transmission of ions, especially at the higher mass range (as you clearly demonstrated in your additional experiments that were requested upon review) and thus impacting on presence/observation of e.g. molecular clusters observed e.g. on VK and DBE plots. Therefore this needs to be taken into account when comparing your data with that of literature (as they used different MS detectors, e.g. FTICRMS and/or acquisition parameters). Therefore, the statement in the line 174 'Thus, we cannot exclude that some new compounds are associated with unrealistic compounds with respect to the atmosphere' needs to be replaced with an appropriate statement reflecting the observed phenomenon. It is also worth adding such statement (based on your figure 9S) to the results and discussion section, e.g. ether after the line 237 or 248 indicating the caveat when comparing your results with the literature data or using their data for your interpretation.

**Here are the changes we have made**

**Line 243-246, page 9 (Results and discussion)**

[revised manuscript text omitted]
. 2 RO2\*  $\rightarrow$  ROOOOR  $\rightarrow$ 2 RO\* + O2, RO2\* + HO2\*  $\rightarrow$  RO\* + \*OH + O2, and RO2\* + NO  $\rightarrow$  RO\* + NO2 followed by alkoxy H-shift (Baldwin and Golden, 1978; Atkinson and Carter, 1991) and peroxidation, RO\*  $\rightarrow$  \*R-HOH; \*R-HOH + O2  $\rightarrow$  \*OOR'OH. The reaction can continue with sequential H-shift and oxygen addition, yielding HOMs via up to six O2 addition in the present study.

(a)

$$C_{10}H_{16} \xrightarrow{-H} C_{10}H_{15} \xrightarrow{+O_{2}} C_{10}H_{15}O_{2} \xrightarrow{RO_{2}} C_{10}H_{15}O_{4}C_{10}H_{15} \xrightarrow{} O_{2} + 2C_{10}H_{15}O \xrightarrow{H=hiff} C_{10}H_{14}OH$$
*Limonene*

$$(R') \xrightarrow{(R')} (I) \xrightarrow{(RO_{2}')} C_{10}H_{15}O_{2} \xrightarrow{RO_{2}} C_{10}H_{15}O_{5} \xrightarrow{H=hiff} C_{10}H_{15}O_{7} \xrightarrow{H=hiff} C_{10}H_{15}O_{9} \xrightarrow{H=hiff} C_{10}H_{15}O_{11}$$

$$(Q'OH)$$

$$C_{10}H_{14}OH \xrightarrow{+O_{2}} C_{10}H_{15}O_{3} \xrightarrow{H=hiff} C_{10}H_{15}O_{5} \xrightarrow{H=hiff} C_{10}H_{15}O_{7} \xrightarrow{H=hiff} C_{10}O_{15}O_{10}O_{10}O_{10}O_{10}O_{10}O_{10}O_{10}O_{10}O_{10}O_{10}O_{10}O_{10}O_{10}O_{10}O_{10}O_{10}O_{10}O_{10}O_{10}O_{10}O_{10}O_{10}O_{10}O_{10}O_{10}O_{10}O_{10}O_{10}O_{10}O_{10}O_{10}O_{10}O_{10}O_{10}O_{10}O_{10}O_{10}O_{10}O_{10}O_{10}O_{10}O_{10}O_{10}O_{10}O_{10}O_{10}O_{10}O_{10}O_{10}O_{10}O_{10}O_{10}O_{10}O_{10}O_{10}O_{10}O_{10}O_{10}O_{10}O_{10}O_{10}O_{10}O_{10}O_{10}O_{10}O_{10}O_{10}O_{10}O_{10}O_{10}O_{10}O_{10}O_{10}O_{10}O_{10}O_{10}O_{10}O_{10}O_{10}O_{10}O_{10}O_{10}O_{10}O_{10}O_{10}O_{10}O_{10}O_{10}O_{10}O_{10}O_{10}O_{10}O_{10}O_{10}O_{10}O_{10}O_{10}O_{10}O_{10}O_{10}O_{10}O_{10}O_{10}O_{10}O_{10}O_{10}O_{10}O_{10}O_{10}O_{10}O_{10}O_{10}O_{10}O_{10}O_{10}O_{10}O_{10}O_{10}O_{10}O_{10}O_{10}O_{10}O_{10}O_{10}O_{10}O_{10}O_{10}O_{10}O_{10}O_{10}O_{10}O_{10}O_{10}O_{10}O_{10}O_{10}O_{10}O_{10}O_{10}O_{10}O_{10}O_{10}O_{10}O_{10}O_{10}O_{10}O_{10}O_{10}O_{10}O_{10}O_{10}O_{10}O_{10}O_{10}O_{10}O_{10}O_{10}O_{10}O_{10}O_{10}O_{10}O_{10}O_{10}O_{10}O_{10}O_{10}O_{10}O_{10}O_{10}O_{10}O_{10}O_{10}O_{10}O_{10}O_{10}O_{10}O_{10}O_{10}O_{10}O_{10}O_{10}O_{10}O_{10}O_{10}O_{10}O_{10}O_{10}O_{10}O_{10}O_{10}O_{10}O_{10}O_{10}O_{10}O_{10}O_{10}O_{10}O_{10}O_{10}O_{10}O_{10}O_{10}O_{10}O_{10}O_{10}O_{10}O_{10}O_{10}O_{10}O_{10}O_{10}O_{10}O_{10}O_{10}O_{10}O_{10}O_{10}O_{10}O_{10}O_{10}O_{10}O_{10}O_{10}O_{10}O_{10}O_{10}O_{10}O_{10}O_{10}O_{10}O_{10}O_{10}O_{10}O_{10}O_{10}O_{10}O_{10}O_{10}O_{10}O_{10}O_{10}O_{10}O_{10}O_{10}O_{10}O_{10}O_{10}O_{10}O_{10}O_{10}O_{10}O_{10}O_{10}O_{10}O_{10}O_{10}O_{10}O_{10}O_{10}O_{10}O_{10}O_{10}O_{10}O_$$

**Figure 7.** Reaction pathways to highly oxygenated products considered in atmospheric chemistry (a) and (b) (Bianchi et al., 2019). Recently extended reaction pathways in combustion (b) (Wang et al., 2017)

The intensity of ions signal decreases with increasing number of O atoms in the  $C_{10}H_{14}O_{2,4,6,8,10}$  (by 5 orders of magnitude) and  $C_{10}H_{14}O_{3,5,7,9,11}$  (by 6 orders of magnitude) products. Nevertheless, the diversity of reaction pathways, associated with the increasing number of chemical compounds, makes it difficult within a population of several hundred chemical compounds to identify all HOMs. Therefore, we have used again the van Krevelen diagram, which allows following the evolution of the oxidation of the first HOMs and to identify them according to definitions that seem to be consensus (Walser et al., 2008;Tu et al., 2016;Nozière et al., 2015;Wang et al., 2017a). To this end, we used the average carbon oxidation state  $OS_c$  which allows distinguishing three regions according to the nature of the functional groups: Region 1 ( $O/C \ge 0.6$  and  $OS_c \ge 0$ ) consists of highly oxygenated and highly oxidized compounds (acids and carbonyls), Region 2 ( $O / C \ge 0.6$  and  $OS_c < 0$ ), consists of highly oxygenated and moderately oxidized compounds (alcohols, esters and peroxides), finally, Region 3 ( $OS_c \ge 0$  and  $H/C \le 1.2$ ) includes compounds with a moderate level of oxygen, but strongly oxidized (Tu et al., 2016).

It can be seen from Figure 3 that autoxidation enhances the development of HOMs, compared to ozonolysis/photooxidation, and that the majority of these new products are found in Regions 1 and 3 of the inset of Figure 3. Thus, further oxidation can go on. We observed products of addition of up to 17 oxygen atoms yielding  $C_{25}H_{32}O_{17}$ .

**Line 516-541, page 24**

These results are in agreement with the previous study by Kourtchev et al. which shows that the evolution of chemical species is mainly dominated by the concentration of OH radical (Kourtchev et al., 2015). The present study has allowed us to highlight autoxidation specific processes, such as formation of KHPs and diketones, occurrence of the Korcek and Waddington reaction mechanisms.

The present results indicate that one should pay more attention to the Korcek and Waddington mechanisms yielding specific products observed here and in previous smog chamber experiments and field measurements.

Extensive oxidation of peroxy radicals yielding HOMs has been considered in atmospheric chemistry, but only recently a third-O2 addition was added to combustion models, showing some influence on ignition modeling (Wang and Sarathy, 2016). Here, limonene oxidation was initiated by reaction with molecular oxygen yielding alkyl radicals which form peroxy radicals by reaction with O2. The oxidation proceeds further by sequential H-shift and O2 addition yielding a wide range of products with odd numbers of O atoms (C10H14O5,7,9,11). Besides, products with even numbers of O atoms were measured in this work ( $C_{10}H_{14}O_{4,6,8,10}$ ). They are expected to come from the oxidation of limonene via the commonly accepted tropospheric oxidation mechanism forming alkoxy radicals, i.e.,  $RO_2^{\bullet} + RO_2^{\bullet} \rightarrow$ ROOOOR  $\rightarrow$  2 RO• + O2. The following sequential H-shift and O2 addition on the alkoxy radicals yielded products of up to six  $O_2$  addition in the present work ( $C_{10}H_{14}O_{10}$ ). Such products have been reported in the previous former atmospheric chemistry studies considered here for comparison (Table 1). Although diverse experimental conditions were used here, in terms of concentration of reactants, temperature, reaction time, analysis conditions, we observed strong similitude in terms of molecular formulae detected in atmospheric and 'combustion' chemistry experiments. Besides, these two routes can produce a pool of OH radicals via decomposition of intermediates, e.g.,  $OOQOOH \rightarrow OH +$ HOOQ'O (KHP) and  $(HOO)_2Q'OO^{\bullet} \rightarrow {}^{\bullet}OH + (HOO)_2Q'O$  (keto dihydroperoxide) for the 'combustion' route and 'OOQOH  $\rightarrow$  'OH + OQ'OH (keto alcool) and HOOQ'(OH)OO'  $\rightarrow$  'OH + HOOQ"(OH)O (keto hydroxy hydroperoxide) for the 'tropospheric' oxidation route. Furthermore, similarly to what has been reported in atmospheric chemistry studies (Witkowski and Gierczak, 2017; Jokinen et al., 2015; Walser et al., 2008; Kundu et al., 2012; Fang et al., 2017; Nørgaard et al., 2013; Bateman et al., 2009;Warscheid and Hoffmann, 2001;Hammes et al., 2019), a wide range of highly oxygenated products were detected, with molecular formula up to  $C_{25}H_{32}O_7$  in the present work.

---

## Author Response (AR4)

Dear Dr Kourtchev,

Thank you for your comments and suggestions. We have taken them into account in the revised manuscript (April 16).

Best regards,

Roland BENOIT

**Comments :** I found a few other statements which are necessary to improve or reduce the ambiguity of the manuscript.

Line 25: replace 'oxygenated products' with 'molecular formulae'
Line 40: abbreviation should be done in the first appearance order, so abbreviate VOC in line 40
Line 47: replace 'α- Pinene' with 'α-pinene'
Line 165: delete repeating sentence
Line 177: remove dots after 'CHO,…)'
Line 178: remove dots after 'CHO,…)'
Line 231: Move HESI settings into the Experiment's section
Line 283: Replace 'chemicals' with 'molecular formulae'
Line 287: Replace 'new chemical formula' with 'additional chemical formulae observed in the current study (also referred as 'new chemical formulae' below)…
Line 325: Replace 'a number of oxygen atoms increasing to 9' with 'up to 9 oxygen atoms'
Line 326: Replace 'globally' with 'mainly' (Please replace 'globally' across the text as this is not the correct choice of word in English).
Line 337: 'global' not sure if I understand this sentence… please rephrase
Line 465: Figure 9, capitalise 'Representation'
Line 469: Remove this sentence as it does not provide any information or value to the results and discussion
Line 478: Replace 'we noticed that' with 'our study suggest that'
Line 478: Complete the sentence as it is not clear similar to what?
Line 492: Remove ' former atmospheric chemistry' from the sentence
Line 513: Replace 'In any case, with' with 'Visualisation tools (e.g. VK diagrams, DBE plots) allowed to differentiate a number of the molecules that are likely related to the experimental conditions used in the current study (e.g. low temperature combustion)'
Line 514: Rephrase 'Among the chemical formulas observed in this work, some had not been reported in the 9 atmosphere-oriented studies considered here for comparison' with 'Among the chemical formulae observed in this work, some have not been reported in the studies considered here for comparison. It should be noted that other factors including experimental conditions (e.g. the use of flow tube reactor vs smog chambers) and/or MS instrument acquisition parameters (e.g. as demonstrated in the SI Figure 9) can be responsible for the observed differences with the compared studies.
I have suggested to add a statement to the introduction and conclusion section to  address the two reviewer's comments on the relevance of the work to the atmospheric chemistry but for some reason it wasn't taken on board. I strongly suggest, adding a statement (at least as suggested above for line 514) to address this concern.

**Here are the changes we have made :**

All comments have been taken into account and changes have been made.

---

## Author Response (AR5)

Dear Dr Kourtchev,

Thank you for your comments and suggestions. We have taken them into account in the revised manuscript (April 16).

Best regards,

Roland BENOIT

**Comments :** Please complete this sentence: 'Our study suggest that in the absence of ozone, the oxidation by the OH˙ radical, common to ozonolysis, gives similar results'. Do you mean similar to previous studies considered in this work? If yes, then it needs to be stated here.

**Here are the changes we have made :**

Our study suggests that in the absence of ozone, the oxidation by the OH˙ radical, common to ozonolysis, gives similar results in terms of observed chemical formulae under the present conditions and those of the studies considered for comparison (Table 1).